# Dual Effect of Immune Cells within Tumour Microenvironment: Pro- and Anti-Tumour Effects and Their Triggers

**DOI:** 10.3390/cancers14071681

**Published:** 2022-03-25

**Authors:** Alicia Cristina Peña-Romero, Esteban Orenes-Piñero

**Affiliations:** Department of Biochemistry and Molecular Biology-A, University of Murcia, 30120 Murcia, Spain; aliciacristina.pr@gmail.com

**Keywords:** immune system, tumour microenvironment, cancer, immune cells, immunosurveillance, immune evasion

## Abstract

**Simple Summary:**

The relationship between the immune system and tumours is currently one of the most studied topics in the field of cancer. Previously, it was thought that a tumour could be malignant on its own; nonetheless, nowadays, we know that it needs the support of different cells and factors, namely, the tumour microenvironment (TME). Among the components that constitute the TME, we find immune cells responsible for supporting the tumour, which play a key role in this tumoural milieu. Although immune cells, under normal conditions, can identify and destroy nascent tumour cells in a process named cancer immunosurveillance, they can be influenced by different factors in the presence of the tumour in a process termed cancer immunoediting. The fact that immune cells can behave as guardians (anti-tumour immunity) or as bystanders or supporters of the tumour (pro-tumour immunity) makes them a “double-edged sword” in the TME.

**Abstract:**

Our body is constantly exposed to pathogens or external threats, but with the immune response that our body can develop, we can fight off and defeat possible attacks or infections. Nevertheless, sometimes this threat comes from an internal factor. Situations such as the existence of a tumour also cause our immune system (IS) to be put on alert. Indeed, the link between immunology and cancer is evident these days, with IS being used as one of the important targets for treating cancer. Our IS is able to eliminate those abnormal or damaged cells found in our body, preventing the uncontrolled proliferation of tumour cells that can lead to cancer. However, in several cases, tumour cells can escape from the IS. It has been observed that immune cells, the extracellular matrix, blood vessels, fat cells and various molecules could support tumour growth and development. Thus, the developing tumour receives structural support, irrigation and energy, among other resources, making its survival and progression possible. All these components that accompany and help the tumour to survive and to grow are called the tumour microenvironment (TME). Given the importance of its presence in the tumour development process, this review will focus on one of the components of the TME: immune cells. Immune cells can support anti-tumour immune response protecting us against tumour cells; nevertheless, they can also behave as pro-tumoural cells, thus promoting tumour progression and survival. In this review, the anti-tumour and pro-tumour immunity of several immune cells will be discussed. In addition, the TME influence on this dual effect will be also analysed.

## 1. Introduction

The National Cancer Institute (NCI) at the National Institutes of Health (NIH) defines the immune system (IS) as “A complex network of cells, tissues, organs, and the substances they make that helps the body fight infections and other diseases”. Immunology is a young science, and it was not until the 1960s when the IS and its functions began to be better understood [1,2]. In 1976, the IS revolutionised modern medicine through vaccination. It was the first time that the IS was manipulated to fight against smallpox, an infectious disease [2]. Edward Jenner, known as “the father of immunology”, was the pioneer in vaccine development, leading to the eradication of smallpox in 1979 [1].

The physical barrier that separates the inside of the body from the outside in vertebrates, comprising the skin and other epithelial surfaces and their narrow joints, protects us from the entry of pathogens. However, there are microorganisms that can cross epithelial barriers/mucosal surfaces in many cases [3]. At this moment, the innate immune system comes into play (mainly made up of granulocytes, mast cells, macrophages, dendritic cells (DCs) and natural killer cells (NKs). However, when this response is not enough, the adaptive immune response comes into play. There are two different adaptive immune responses: the humoral immunity (mediated by B lymphocytes responsible for producing antibodies) and cellular immunity (mediated by T lymphocytes responsible for lysing the infected cell) [2].

The relationship between IS and cancer was not considered and studied until the 20th century (1909), when Paul Ehrlich suggested the importance of an individual’s IS for protection against cancer, thus including the term “immunosurveillance” [4]. Later, in the 1950s, Burnet and Thomas included the concept of “cancer immunosurveillance” [5,6]. More recently, different components of the IS have been used as targets to treat inflammatory diseases and cancer [2]. In the 21st century (2002), interests in the relationship between IS and cancer grew, and Robert Schreiber described the concept of “cancer immunoediting” for the first time [7].

The term “tumour microenvironment” (TME) was firstly described in 1889, when Stephen Paget, known as the conceptual father of the tumour microenvironment and tumour progression, developed this term with his seed and soil theory. The TME is not only composed of tumour cells, but also of stromal cells (non-tumour cells), which have a major influence on tumourigenesis, pathogenesis and tumour progression, as well as metastasis [8,9].

One type of non-tumour cells found in the TME is immune cells. On the one hand, when they adopt the anti-tumourigenic phenotype they can contribute to the immunosurveillance, preventing tumour progression, but on the other hand, immune cells can adopt the pro-tumourigenic phenotype by TME influence, allowing the tumour escape, and even supporting the TME, thereby enhancing tumour progression [7,10,11]. The aim of this review is to thoroughly discuss the relationship between immune cells and TME in cancer.

## 2. Methods

Published data from this review were identified by searching and selecting publications in the PubMed, Nature Reviews/Cancer, MEDLINE, Google Scholar, Semantic Scholar and Elsevier database and by searching reference lists of relevant articles, reviews and books. Different scientific journals were also employed. Some basic definitions and/or concepts on the subject were obtained through the National Cancer Institute at the NIH (National Institutes of Health) and Medlineplus.gov (NIH—United States National Library of Medicine) website. A four-step approach was used. Firstly, an overview of the history of the science of immunology and how the IS works under normal conditions was carried out through a combined keyword search: “history”, “hypothesis”, “background of immune system”, “immunology”, “immune system”, “haematopoiesis”, “immune response”, “cellular immunology”, “molecular immunology”. Secondly, a general search regarding cancer disease and tumour microenvironment was addressed by combining the following keywords: “hallmarks of cancer”, “immune system”, “behaviour”, “tumour”, “cancer”, “relationship”, “tumour microenvironment”. The third-step approach was to highlight the cancer immunoediting by searching mainly: “cancer immunoediting”, “tumour escape” and “cancer immunosurveillance”. Lastly, the dual effect of immune cells in tumour microenvironment and how this latter influenced them was written through the search for words such as: “(type of immune cell) anti-tumour immunity”, “(type of immune cell) pro-tumour immunity”), “(type of immune cell) in tumour microenvironment, “(type of immune cell) dual effect”), and tumour-microenvironment-derived factors.

Reference lists of all selected articles and review articles about IS and TME were also reviewed for other relevant articles.

## 3. Overview of Immune System

### 3.1. Immune System in Health

The IS consists of cells, organs and tissues and substances produced during their activation. It comprises different cell populations: leukocytes (also called white blood cells) divided into granulocytes (neutrophils, eosinophils and basophils), monocytes and lymphocytes (T and B lymphocytes); erythrocytes; platelets; mast cells; DCs and innate lymphoid cells (ILCs). All of these cells originate in the bone marrow from hematopoietic stem cells (HSC) in a process called haematopoiesis (Figure 1). Regarding the organs and tissues that integrate the IS, primary and secondary lymphoid organs can be found. On the one hand, the primary lymphoid organs (also called central lymphoid organs) comprise the bone marrow and the thymus and are thus called because this is where the antigen- or injury-dependent maturation of lymphocytes (lymphopoiesis) takes place. On the other hand, the secondary lymphoid organs (also called peripheral lymphoid organs) are where the previously matured lymphocytes interact with the antigen-presenting cells (APCs) and thus, along with the antigen, thereby develop an immune response that spreads throughout the body.

In the IS, several compounds make it possible for the correct and timely activation of the immune system to take place. These include chemical substances, adhesion molecules (selectins, integrins, cadherins (calcium-dependent adherins, addresins) and blood proteins such as antibodies, complement proteins and cytokines (interferons, chemokines) [1,2,12].

### 3.2. Immune Response

All the components of the immune system, molecular and cellular, are coordinated and regulated in order to carry out a proper IR [2,13,14]. The presence, but also the location and function of cells of both immunity (innate and adaptive) is required for the immune response to take place. Immune cells are distributed throughout our body [2,13].

It is important to note that the innate and adaptive IR are interlinked to each other, i.e., they work in a correlated manner [14,15]. This connection between innate and adaptive is through APCs activating TLRs (Toll-like receptors) or mediated by fragment crystallisable (Fc) receptors [14,16,17]. These APCs include macrophages, B lymphocytes and DCs, the latter being the most effective to activate naïve T cells [2].

#### 3.2.1. Innate (Natural or Native) Immunity

The word “innate” derives from the Latin word “innātus” and means “inborn”. Thus, this is called innate immunity because we are born with it, and it is capable of triggering an immune response without the need to have been in contact with the pathogen beforehand. It is called an early defense because it is quick, i.e., it is activated in a few hours or days after the infection has occurred in our body; however, this, in turn, makes it non-specific. In addition, does not change in intensity with exposure [2]. The innate IR is designed to prevent infection, to eliminate invader pathogen and to stimulate the acquired immune response [18]. It is of vital importance for survival, for this reason it is also found in the simplest animals [19].

Concerning their features, they are traditionally thought to be non-specific [20]; nevertheless, this concept has been challenged by the discovery of pattern recognition receptors (PRRs) [21], which are activated in presence of pathogen-associated molecular patterns (PAMPs) and molecules produced by damaged host cells (damage-associated molecular patterns (DAMPs) [2].

Another characteristic is its immune tolerance against self-antigens, i.e., no IR is triggered against self-antigens in the organism; in some cases, this process may not work properly and give rise to autoimmunity [2,22,23].

Innate immunity mainly comprises the following components: (1) physicochemical: epithelial barriers (physical barrier such as tight junctions in the skin (that is the largest), and mucous membrane surfaces; chemical barrier such as pH and antimicrobial molecules and biological barrier (human intestinal microbial flora); (2) humoral: blood proteins, such as the elements of the complement system and other mediators of inflammation such as cytokines, C-reactive protein (CRP) or proteolytic enzymes; (3) cellular: granulocytes (neutrophils, eosinophils and basophils); DCs (involved in the regulation of the immune response but also in phagocytosis); macrophages; mast cells; NK cells and other ILCs [2,24,25,26].

Skin and other epithelial surfaces (physical barrier) protect the vertebrate body from the possible entry of pathogens; nevertheless, this is not always sufficient, since microorganisms can cross epithelial barriers/mucosal surfaces in many cases [3], being, thus, of vital importance, the activation of innate IR [2]. At this point, mast cells, basophils and eosinophils are the first cells that comes into play since they can be found in epithelial surfaces [2,12]. In addition, Langerhans cells (LCs), a DCs subset, are a population of mononuclear phagocytes that reside in the epidermis and are associated with the acquisition of DC-like functions (antigen-presenting function) in dangerous situations [27,28,29]. Each microorganism is characterised by different PAMPs that will be recognised by different PRRs, present in APCs, as in the case of DCs. Among them, TLRs, are the best characterised receptors that can trigger activation of adaptive IR [15,30].

Soluble receptors, such as components of the complement system also play an important role in the onset of phagocytosis and also, sometimes, destroy the pathogen directly [3]. It consists of several plasma proteins that work together to opsonise pathogens, to promote the recruitment of phagocytes to the site of infection and in some cases, to directly kill the microbes by lysis. Its activation can occur by three different processes: classical, alternative and lectin pathway [1,2,12,31,32].

In this context, IR begin with the migration of neutrophils (polymorphonuclear neutrophils (PMN), characterised by chemotaxis), to the site of inflammation, followed by macrophages (considered phagocytic cells) being, thus, the first to respond to a common pathogen, and accordingly, the first line of defence in the innate IR [1,3].

Neutrophils and macrophages engulf and destroy particles, whereas NK cells (cytotoxic cells) kill parasites and infected cells and produce interferon gamma (IFN-γ), which activates macrophages to destroy phagocytosed microbes [2,33]. The cytokines, which regulate and coordinate the several interactions carried out in the IR, in this phase, are produced mainly by macrophages, NK cells, and by other non-immune cells such as fibroblasts and endothelial cells [34].

Later, others activated macrophages will bind monocytes and dendritic cells [3] and will attract neutrophils. The DC (known as sentinels of the immune system [35,36]) capture and process the antigens and then transport them to the lymph nodes, where they are presented for recognition by T lymphocytes [3].

DCs rapidly and efficiently detect invading microbes owing to their location (tissues, lymphoid and non-lymphoid organs, as well as circulating in afferent lymph and peripheral blood) and their expression of numerous PRRs for PAMPs and DAMPs [2,37].

Effective activation of T lymphocytes by DCs requires several consecutive signals and their interactions depend on the degree of maturation and lineage of DCs [38]. DCs can activate both CD4^+^ T lymphocytes (CD4^+^ T cells or Th) and CD8^+^ T lymphocytes (cytotoxic T lymphocytes or CTLs) by antigen presentation via MHC-II and MHC-I (major histocompatibility complex), respectively (cross-presentation) [35].

#### 3.2.2. Adaptive (Specific or Acquired) Immunity

Once the innate immune response has been activated, the adaptive specific immune response develops (needs days or weeks), if necessary (depending on the injury or as a consequence of the failure of the innate immune response) [16]. It is triggered in the presence of the pathogen, and it is prepared to recognise it in case of new attacks or infections, thus avoiding being re-infected by the same pathogen (immunological memory). In addition, unlike innate immunity, it does change in intensity after repeated exposure to the pathogen [1,2,39]. Adaptive immunity, unlike innate immunity, is not found in all animals, being characteristic of higher animals [12].

In relation to its characteristics, it has been shown to be specific in the recognition of microbial and non-microbial antigens by gene recombination of T-cell receptor (TCR) and B-cell receptor (BCR) [2]. Immunological memory of the adaptive immune response is not limited, but it is a hallmark of this type of immune response. The existence of this immunological memory, characterised by both, higher magnitude/kinetics and specificity than the innate IR, allow the development of a faster and more specific response [2,40], owing to the ability of the lymphocytes to remember a pathogen, preparing for a re-exposure to it. Immunological memory is carried out by both, T and B cells and it is owing to antigen hyper-reactivity [41]. Additionally, it is a diverse response, as receptor genes are formed by somatic recombination of gene segments in lymphocytes in a process called: V(D)J recombination [2,42].

As in innate IR, under normal conditions, there is an immunological tolerance by self-antigens [2,12], thus, tolerance and immunity have to be balanced. Thus, there must be a balance mediated by survival, activation and expansion of the lymphocyte clones (immunity) and inhibition and elimination of the clones (tolerance) [43]. The main components of this type of immunity are lymphocytes (white blood cells). Lymphocytes in epithelia and antibodies (blood proteins) secreted by epithelial surfaces constitute the cellular and chemical barriers of adaptive immunity [2].

During the development of adaptive IR, clonal selection (specificity of lymphocytes for an epitope of an antigen) and the clonal expansion (stimulation, proliferation and differentiation of lymphocytes) take place. Adaptive IR occurs in the secondary lymphatic organs (lymph nodes, spleen, lymphoid tissue associated with mucous membranes) where the lymphocytes recognise the pathogens previously presented by the DCs. These lymphocytes activate and multiply (clonal expansion), thus initiating the adaptive IR. Accordingly, the differentiation of lymphocytes into effector cells capable of eliminating the antigen (the presence of macrophages and neutrophils is needed) and memory cells takes place. Once the IR is complete, the lymphocytes are inactivated, and the effector cells die. The memory T lymphocytes remain (responsible for immunological memory) [2].

Two types of adaptive immunity can be found: humoral and cellular. The humoral immunity is mediated by activated B lymphocytes (plasma cells) that generate large amounts of antibodies (immunoglobulins M (IgM), IgG, IgA, IgE) each with a function. They complete their differentiation and become activated in response to antigen-dependent T-cells [2,44,45]. B cells are responsible for recognising the antigen, neutralising it and attacking it to be then eliminated by phagocytes and the complement system. Although B cells can be activated directly by non-protein antigens (lipids, polysaccharides), their response to protein antigens requires the help of CD4^+^ T cells (also called T helper lymphocytes—Th) [2].

Memory B cells longevity and rapid and robust responses to antigen re-exposure are the basis of vaccine success [46].

In contrast, the cellular immunity is mediated by T lymphocytes, which come into play when microorganisms, despite being phagocytosed, cannot be eliminated. Thus, T cells proceed to the destruction of these microbes and the possible infected cells, (viruses) recognising the peptides of foreign proteins presented by the MHC [2,47].

## 4. Cancer Disease and Tumour Microenvironment

### 4.1. Cancer Pathophysiology—Hallmarks of Cancer

NCI defines tumour as follows, “An abnormal mass of tissue that forms when cells grow and divide more than they should or do not die when they should. Tumours may be benign (not cancer) or malignant (cancer)” [48]. In the 20th century, it could be proved that tumours are created by cells that have lost the ability to assemble and create tissues of normal form and function [49]; thus, it is characterised by the accumulation of genetic mutations and the loss of normal cellular regulatory functions [50].

As mentioned above, tumours can be classified as benign or malignant: the difference between them is the cell growth speed, the existence of invasion into the surrounding tissues and the existence of metastasis [49,51,52]. The importance of this uncontrolled cell division as a cause of the development of many human cancers was suggested in 1990 [53].

According to the World Health Organization (WHO), cancer is a multifactorial disease, due to the combined effect of genetic and environmental factors [54]. Cancers are increasingly heterogeneous as they progress; even after malignant transformation a cancer remains dynamic and continues to evolve. Hence, cancer is considered to be a dynamic disease, resulting in tumour heterogeneity [49].

The concept of tumour heterogeneity can be broadly divided into inter- and intra-tumoural heterogeneity. On the one hand, inter-tumoural heterogeneity (also called interpatient heterogeneity) refers to heterogeneity between patients harbouring tumours of the same histological type. The reason for this difference is due to patient-specific factors, such as germline genetic variations, differences in somatic mutation profile, and environmental factors. On the other hand, intra-tumoural heterogeneity refers to genetic heterogeneity among the tumour cells of a single patient. There are two types of intra-tumoural heterogeneity: a) spatial heterogeneity: in different disease sites (intermetastatic heterogeneity) or within a single disease site or tumour (intratumoural and intrametastatic heterogeneity) and b) temporal heterogeneity: genetic diversity produced over time [55,56,57]. Spatial heterogeneity is a fundamental feature of the tumour microenvironment, as has been documented in several cancers such as melanoma [58], breast cancer [59,60] or giloblastoma [61], among others. Causes of intratumoural heterogeneity include genomic instability and clonal evolution [56]. Genomic instability triggers mutations between genes and different domains within the same gene [62]. Clonal evolution, suggested by Peter Nowell in 1976 [63], originated from somatic cellular mutations [56].

The complexity and variability of this disease makes it difficult to understand; for this reason, in 2000, the concept of “hallmarks of cancer” was proposed to constitute an organising principle for rationalising the complexities of neoplastic disease and hence understand their biology [64]. These hallmarks allow cancer cells to survive, proliferate, and disseminate [65]. In 2000, hallmarks of cancer involved six biological capabilities acquired by cancer cells during the long process of tumour development and malignant progression [64]; nevertheless, in 2011, this increased to eight capabilities and two enabling characteristics [65,66]. On the one hand, the eight capacities that constitute the “hallmarks of cancer” are: (1) sustaining proliferative signaling, (2) evading growth suppressors, (3) resisting cell death, (4) enabling replicative immortality, (5) inducing angiogenesis, (6) activating invasion and metastasis, (7) deregulating cellular energetics and metabolism and 8) avoiding immune destruction. On the other hand, the two enabling capabilities are: (1) genome instability and the consequent mutation of hallmark-enabling genes and (2) tumour-promoting immune cell infiltration [66].

The evolution of normal cells towards a neoplastic state involves a succession of these distinctive capabilities [65]. This process carries with it heterotypic interactions between multiple cell types that populate the TME [66].

### 4.2. Tumour Microenvironment

For a long time, nearly all studies were focused exclusively on the individual tumour cells [67,68], and it was thought that tumours were insular masses of proliferating cancer cells [65]; however, in the 19th century, Paget’s description of TME by seed and soil theory confirmed that tumours contain various noncancerous cells [69]. Nonetheless, Paget’s concept was not revisited until mid-seventies of the 20th century [8,70,71,72]. Accordingly, in the last 25 years, tumours have been studied as a complex organ, showing that entirety of the tumour components (neoplastic cells) are not malignant by themselves [67] but they need the contribution of tumour stroma (non-neoplastic cells) to promote the tumourigenesis [65], including a persistent inflammation. Indeed, in their initial stage, tumour cells are able to instruct the TME for the formation of blood vessels to recruit nutritive factors, cell-derived vesicles and immune cells, thus contributing to the development of many hallmarks of cancer [65,73].

Accordingly, the tumour stroma, constituting the TME along with tumour cells, comprise non-tumour cells that surround the tumour cells present in the TME [74,75], including basement membrane, extracellular matrix (ECM), fibroblasts, various subtypes of cancer-associated fibroblasts (CAF), angiogenic vascular cells (AVC), endothelial cells, supporting perycites, glial, smooth muscle, epithelial, fat cells, and infiltrating immune cells (IIC) [9,76,77]. These latter are the bulk of the stromal components along with CAF, AVC, endothelial cells and supporting pericytes [9]. In addition, the TME also harbors cytokines, chemokines, growth factors and antibodies [78,79].

How TME affects tumourigenesis, cancer progression and metastasis is increasingly being studied [65,80,81,82]. Indeed, in 2006, the NCI (Bethesda, Washington D.C., MD, USA) launched the Tumour Microenvironment Network (TMEN), whose main goal was to study the mechanisms of tumour-host interactions in human cancer. TMEN investigators in the 2006–2011 Consortium firmly established the influence of tumour-stroma interactions on tumour progression, among others [83]. According to the NCI, “A tumour can change its microenvironment, and the microenvironment can affect how a tumour grows and spreads” [48].

In this context, the recruited immune cells can infiltrate the TME, promoting both, the anti-tumour immunity and the pro-tumour immunity, and can co-evolve from local disease to become migratory, invasive, and angiogenic [66,76,83]. All these features give rise to tumour growth, the evasion of immune surveillance and the resistance to immunotherapies [65,84,85,86]. It is worth noting that cells can adopt either anti-tumoural or pro-tumoural phenotype and can even be innocent bystander cells, as in the case of tumour-associated mast cells (TAMCs) or B cells [87].

The TME plays a pivotal role modulating immune response in the setting of cancer. Tumour cells and their microenvironment often produce numerous immunomodulatory molecules that can negatively (inhibitory factors) or positively (activating factors) influence the functions of immune cells [88,89,90]. Hence, the TME is capable of switching the immune response from a tumour-destructive profile to a tumour-promoting profile, depending on the composition of the aforementioned TME [91,92]. Accordingly, TME soluble mediators (cytokines, chemokines, angiogenic, lymphangiogenic and growth factors) and cellular receptors play a key role in IR. For instance, in human prostate cancer it has been reported that the prognostic role of TAMCs depends on the tumour stage and on the mast cells’ location within the tumour tissue [93,94,95]. Additionally, tumour type and stage [96], tumour-associated neutrophils (TANs) [97] and eosinophils [98], among others, can modify the IR in a positive (anti-tumour) or a negative manner (pro-tumour).

Regarding tumour stage, although the stroma is composed mostly of cells that possess certain tumour-suppressing abilities, the stroma could lose this capacity during malignancy and, therefore, could play crucial roles in tumourigenesis, cancer progression, metastasis, and therapy resistance [74,76,77]. Thereby, stromal cells in the TME dynamically interact with cancer cells, and can either compete or cooperate with them by altering their genotype and phenotype [99]. Nevertheless, this cannot be clearly stated, being unclear and controversial. For instance, although mast cells density has been correlated either with good or poor prognosis, depending on the tumour type and stage [98], in breast cancer, it was reported that in some cases the good prognosis by mast cells infiltration, was independent of age, tumour grade, and molecular cancer subtype [100,101,102].

It is important to remark that tumour heterogeneity (intertumoural and intratumoural heterogeneity) may also influence the TME behaviour. In this context, tumour immunogenicity varies greatly between types of cancer and between different individuals with the same type of cancer [98]. Of note, in addition to interactions between tumour cells and stromal components of the TME, cell-autonomous genetic and epigenetic changes are also involved [65,80]; thus, they may form a favourable environment for the development of cancer [103]. This way, cancer cells are able to proliferate autonomously, resist cell death, evade the immune system, replicate immortally and invade other tissues, triggering metastasis [65].

## 5. Cancer Immunoediting—From Immunosurveillance to Immune Evasion/Tumour Escape

The term “immunosurveillance” suggests that the immune system can identify and destroy early tumours in the absence of therapeutics [4,104]. Indeed, it is well known that a single CD8^+^ T cell is capable of carrying out from 1 up to 20 kills in a day [105,106]. In support of this, it has been accepted the concept of “cancer-immunity cycle”, harbouring the seven steps performed by the IS against the tumour, demonstrating that IS is capable of selecting and eliminating tumour cells, highlighting the presence of APCs and activated T cells, responsible for attacking tumour cells [107].

Nevertheless, immune cells may loss this ability and be subjected to conversion from anti-tumour cells to immunosuppressor cells and even pro-tumoural cells [104]. For this, tumour cells carry out a set of processes that take place either independently or in sequence known as cancer immunoediting. Thus, the processes that take place during the switch from anti-tumour to immunosuppressive role are: (1) Elimination, (2) Equilibrium and (3) Escape [108]. Elimination phase may be referred to as protection, since, in this phase, innate and adaptive immunity can eradicate the developing tumour and protect the body from tumour formation by eliminating the transformed cells before they become clinically evident [10,108]. If this process is not successful, malignant cells may not be completely eradicated but instead enter into an equilibrium phase [108]. In this phase, a persistence exists, that is, tumour cells variants (clinically undetectable) that survive to the elimination phase, continue to expand, under the supervision of the IS, which is able to limit their growth by selecting for immunologically resistant variants, in addition to edit tumour cell immunogenicity. This phase is the longest of the three phrases, as it may occur over a period of years [104]. Notwithstanding, this state of coexistence that arise between tumour cells and immune cells in equilibrium phase may be interrupted by the victory of the tumour cells against the IS supervision to which they have been subjected, leading to what is known as tumour escape (immune evasion). This phase give rise to tumour progression given that additional tumour cells variants, edited and of poor immunogenicity, evade the immune system through direct and/or indirect mechanisms [109] and begin to grow progressively in an immunologically unrestrained manner, leading to metastasis. Clinically detectable cancers often represent this escape phase [7,104,110].

There are several immune evasion mechanisms that can occur in the scape phase, such as defective differentiation and function of APCs; the loss of expression of MHC-I molecules [111,112]; T cell dysfunction within the TME or upregulation expression of cytotoxic T cell inhibitory ligands [113], including programmed cell death-ligand 1 (PD-L1); cytotoxic T-lymphocyte-associated protein 4 (CTLA-4); lymphocyte activation gene-3 (LAG-3); T cell immunoglobulin and mucin domain-containing protein 3 (TIM-3), and 4-1BB [114,115,116,117]; or intratumoural T cell exhaustion (due to chronic antigenic stimulation). It is worth noting that among the most recent studies, the IFN-alpha/beta Receptor 1 (IFNAR1) degradation [118], a notion that was supported by a previous study [119], through the correlation between Type I interferon (IFN-I) downregulation and granulocytic myeloid-derived suppressor cells (G-MDSCs) immunosuppression, has been considered a new mechanism for tumour evasion.

In general terms, it could be said that tumour cells evade immunosurveillance by editing their microenvironment, including immune cells and their neighbouring normal cells [110], reducing the immunogenicity of the tumour cells (by autoantigens expressed in tumour cells) or altering the immune cells functions by TME [120,121].

Importantly, genetic and epigenetic changes have been involved in TME behaviour [65,80]; thus, these changes can contribute to the tumour escape, conferring resistance to immune recognition and elimination [108]. Indeed, the importance of the loss of various tumour suppressor genes (cancer- and tissue-specific) has been studied, managing the proliferation of cancers in vivo in the presence of the adaptive immune system [122]. Accordingly, cancer immunoediting can be defined as a dynamic process wherein immunity functions act not only as an extrinsic tumour suppressor but also shaping tumour immunogenicity [104]. In this regard, the crucial role of cancer immunoediting has been reported in several cancers such as lung cancer [110,123,124], colorectal cancer (CRC) [125,126], melanoma [127], breast tumour [128], prostate [129] and multiple myeloma [130], among others.

## 6. Dual Effect of Immune Cells in Cancer

In this section, how TME influences both anti-tumour and pro-tumour immunity of immune cells will be discussed. Among the factors involved, tumour-derived cytokines, chemokines and even metabolic conditions (pH, oxygen levels and nutrients) can influence in determining immune cell function in several cancers. Hence, depending on the TME, immune cells will secrete different factors accordingly, determining its anti-tumour or pro-tumoural role [98,131] (Figure 2).

### 6.1. Immune Innate Response

#### 6.1.1. Granulocytes (Neutrophils, Eosinophils and Basophils)

▪Anti-tumour immunity

The presence of neutrophils in TME is well-known [132,133,134]; nonetheless, in recent years, the function of neutrophils in cancer is controversial due to the fact that their behaviour depends on the context [97] (N1 anti-tumoural or N2 pro-tumoural phenotype) [135,136]. Notwithstanding, it should be highlighted that this neutrophil polarisation (N1, N2) is a kind of artificial classification to the fore, as the phenotype they adopt in the real TME is more complex and cannot be simply classified into binary states. Indeed, the mechanism involved in neutrophil polarisation is unclear, suggesting that reprogramming of mature neutrophils may be triggered by external stimuli or that programming of the defined phenotypes may arise in the bone marrow. Furthermore, the distinction between these two phenotypes has also been defined with respect to the balance of the cytokines transforming growth factor beta (TGF-β) and interferon beta (IFN-β) in the TME [137,138]. In this way, given the plasticity of neutrophils and the complex behaviour they can adopt, this classification cannot be determinative in the real TME. In this context, more recently, it has been observed that the level of reactive oxygen species/reactive nitrogen species (ROS/RNS) production by neutrophils modulate phagocytosis, secretion, gene expression, and apoptosis [139], and thus can dictate their pro- or anti-tumour behaviour in the TME [140]. Its anti-tumoural function was reported two decades ago causing tumour rejection of granulocyte colony stimulating factor (G-CSF)-producing colon cancer cells transplanted into mice [141]; nevertheless, a few years later its dual effect was demonstrated, since neutrophil depletion in mice bearing transplantable tumours showed a reduced tumour growth [142,143].

The TME behaviour “help” to neutrophils recruitment, since, the behaviour of tumour cells and the surrounding microenvironment promote their alertness, with consequent recruitment. Factors implicated in promoting their role as anti-tumour cells in the TME, leading to limit tumour growth, include hypoxia-induced C-X-C chemokine ligand 1 (CXCL1), CXCL2 and CXCL5 expression [144]; proto-oncogene c-met (MET) [145]; hydrogen peroxide (recruit leukocytes to transformed cells like in wounds) [146]; chemokines and cytokines signals (many of them are also important in acute wounding). Concerning this latter, include, (but are not limited to): C-X-C Motif Chemokine Receptor 1 (CXCR1) and CXCR2 (promote the recruitment of TANs to cancer) [147]; CXCL8, CXCL6 and CXCL5 [148]; Tumour Necrosis Factor alpha (TNF-α) [149]; Gamma-delta T cells (γδ T cells)-derived IL-17 (Interleukin-17) [150,151]. The latter two have been reported in the mouse skin carcinogenesis model [149] and in metastatic breast cancer [150,151], respectively. In addition, neutrophil inducible nitric oxide synthase (iNOS) expression also has a beneficial role, since it is involved in cytotoxicity activity exerted on cancer cells [152]. Moreover, in early-stage human lung cancer, TANs promoted T cell cytotoxic function through the production of OX-40 ligand (OX-40L) and 4-1BBL (a member of the TNF) co-stimulatory molecules [153] (Table 1).

Regarding eosinophils, their role in cancer has largely been overlooked [98]. The presence and increase in peripheral eosinophils in patients with cancer was reported more than 120 years ago [175]. Similarly to neutrophils, they might be polarised to an anti-tumourigenic or pro-tumourigenic phenotype in response to stimuli present in the TME. Moreover, due to its plasticity, they can also remodel the TME by releasing of extracellular vesicles [98]. For this reason, although in 1980s, tumour-infiltrating eosinophils were associated to good prognosis in human gastric cancers [176] and also in melanoma, colon cancer, oral squamous cell carcinoma (OSCC) and lung cancer [177]; its role as pro-tumourigenic was also considered, as reported in cervical carcinoma patients [178]. Nowadays, it is well-known that eosinophils can comprise a substantial proportion of the immune infiltrate of the TME [179] and be involved in new therapeutic strategies [180], as well as considered potential prognostic role in human cancers [98].

TME-factors expressing involved in their recruitment, include cytokines (IL-5, IL-10, IL-12) [181]; eotaxins activating C-C chemokine receptor type 3 (CCR3) (eotaxin–CCR3 axis), receptor that is highly expressed on eosinophils; chemokines (chemoattractant) such as C-C Motif Chemokine Ligand 3 (CCL3), CCL5 (also known as RANTES) involving CCR1; DAMPs such as high mobility group box 1 (HMGB1) and IL-33 that is capable of inducing Th2 cell immune response and microbiota (direct eosinophil–bacteria interactions) [181].

They can contribute to anti-tumour activity in a direct way (by granule proteins, granzyme A, involved in strong cytotoxic activity) [182] as reported in human colon carcinoma cells [154,155] (Table 1) or in an indirect way with CCL5, CXCL9 and CXCL10 expression that contribute to the recruitment of tumour-reactive CD8^+^ T cells, as reported in mouse models of melanoma [177]. Acting as APC can also promote a direct anti-tumour activity; however, it has to be demonstrated if its role, similar to APCs, can carry out also within the TME [183]. It should be emphasised that eosinophils and mast-cells can modulated each other, in a bidirectional way, which could be relevant in TME [98].

Lastly, concerning basophils, although they account for less than 1% of peripheral leukocytes, and only its role in allergic inflammation and parasite infection has been considered, recent studies have now highlighted their role in cancers [184,185]. As a matter of fact, in some studies, significant numbers of basophils were found in autologous breast cancer tissue [186]. As for neutrophils and eosinophils, also basophils are recruited under influence of TME [187,188], through different mechanisms (closely associated with the secretion of chemokines CCL7/MCP3 (monocyte-chemotactic protein 3) by M2 macrophages, and T cell-derived IL-3.

Several cytokines are secreted and released by basophils: IL-4 (reduced the M2 macrophage polarisation in colon cancer-bearing mice leading to further survival) [189], IL-13 (contributed to activate innate immune responses in melanoma and fibrosarcoma-bearing mice resulting in inhibiting of tumour growth [190], CCL3 [157] and CCL4 [191] (both involved in CD8^+^ T cell infiltration in murine colon cancer) [157] (Table 1). In addition, IL-4 and IL-3 are also crucial during Th2 immune response [189,190].

▪Pro-tumour immunity

TME is often termed as a “wound that does not heal” due to its persistent inflammation [192]. This is a pivotal feature in initiating tumourigenesis, involving the damaged specific tissues and the required presence of neutrophils to be executed [152]. For this reason, there are common mechanisms that mediate neutrophils recruitment to wounds and cancer [97]. Indeed, over the past twenty years it has been well-known that cell mutations are required but not sufficient for the development of tumourigenesis, owing to the required neutrophils [193] which contribute to chronic inflammation and tissue damage when upregulated [97]. Thus, they have been recently linked to both, proliferation and invasion of cancer cells [97].

The pro-tumour function of neutrophils can be implicated in tumour initiation by prostaglandin E2 (PGE2) expression [194], the promotion of tumour growth by TNF-induced IL-17-producing CD4^+^ T cells [195] and by matrix metalloproteinase 8 or 9 (MMP8-MMP9 that remodel the ECM and induce angiogenesis [196]. They can also be associated with the progression of tumour growth, by IL-1 receptor antagonist (IL-1RA) [197] and by the neutrophil extracellular traps (NETs) implicated in senescent cancer cells conversion into proliferating cancer cells [198]. Additionally, they are also implicated in metastasis via coiled-coil domain containing protein 25 (CCDC25) [199,200]; proliferation, by transfer of neutrophil elastase (NE) to cancer cells [201] and immunosuppression, by TGF-β and GCS-F signalling in neutrophils [152].

This way, circulating tumour cells and white blood cells can be clustered determining cancer progression [202,203]. As a matter of fact, in a study carried out on human and mouse breast cancer, it was reported that the majority of these white blood cells clustered with circulating tumour cells were neutrophils. This clustering promoted cell cycle progression and with it, metastasis, evidencing their pro-tumoural activity. Neutrophils, under these conditions, expressed cytokines such as TNF-α, Oncostatin M, IL-1β and IL-6, promoting tumour growth. However, although the overexpression of factors such as G-CSF favoured circulating tumour and white blood cells cluster, the inhibition of vascular cell adhesion molecule 1 (VCAM1) or lymphocyte antigen 6 complex locus G6D (Ly-6G) prevented this association [204].

Another mechanism contributing to cancer metastasis is through the NET-DNA binding to CCDC25 on tumour cells, as reported in patients with breast cancer and patients with colon cancer [199,200]. One of these studies reported that NET-DNA induced migration, adhesion and proliferation of tumour cells via interaction with CCDC25 [199].

A recent study also highlighted the pro-tumour activity that neutrophils can acquire. In this study, Fatty acid transporter protein 2 (FATP2) was found to be a factor involved in promoting tumour growth by neutrophils. The immunosuppressive function of PMN myeloid derived suppressor cells (PMN-MDSC), which pathologically activated neutrophils, was regulated by FATP2, showing the highest expression in tumour site, being able to demonstrate its suppressor activity mainly through the regulation of the accumulation of arachidonic acid and subsequent synthesis of PGE2. Indeed, lipid accumulation of PMN-MDSC from cancer patients (head and neck, lung, or breast cancers) is higher than PMN from healthy donors and the abrogation of PMN-MDSC suppressor activity occurred after FATP2 deletion [205].

Recently, another study showed that neutrophils promoted tumour cell proliferation in human gastric cancer. In this case, the involved pathway was TNF-α-B7-H2-IL-17A. Tumour-derived TNF-α promoted neutrophil activation and neutrophil B7-H2 expression through the extracellular signal-regulated kinase-nuclear factor kappa light chain enhancer of activated B cells (ERK-NF-κB) and the Th producing-IL-17A pathway which was, in turn, polarised in a B7-H2-dependent manner, resulting in pro-tumourigenic roles [206].

On the other hand, iNOS production, apart from showing anti-tumour activity, can acts by suppressing anti-tumour immunity as well, suppressing effector CD8^+^ T cell along with arginase 1 (ARG1) and resulting in tumour progression [152]. An indirect pro-tumoural function of neutrophils was demonstrated, since the intraluminal NK cell-mediated tumour cell clearance could be suppressed by them [207].

It should be emphasised that, although CXCL5 and CXCR2 were mentioned, similar to anti-tumour chemokines, the CXCL5/CXCR2 axis, instead, can contribute to tumour invasion in hepatocellular carcinoma. In addition, Oncostatin M (member of the IL-6) contributes to tumour invasion as reported in human breast cancer (involved in vascular endothelial growth factor F (VEGF) induction) and mouse lung cancer (by accumulation of M2 macrophages) [97] (Table 1).

On the other hand, the pro-tumourigenic activities of eosinophils comprise tumour growth and reduction in the tumour cells death [208], among others (Table 1). In order to do this, different mechanisms have been proposed: tumour growth can be caused through eosinophil-derived CCL22, by thymic stromal lymphopoietin (TSLP) that induces angiogenesis via VEGFA and by eosinophils-derived epidermal growth factor (EGF) along with TGF-β. Immunosuppression was observed by eosinophils-derived indoleamine 2,3-dioxygenase (IDO) (inhibition of effector T cells) and by an increasing of IL-4 and IL-13-expression by eosinophils via TSLP leading to conversion of macrophages into tumour-promoting activation state. Lastly, MMP9 can be also pro-tumour along with MMP2, and both of them are involved in metastasis [181]. In diseases such as human cervical cancer [178] eosinophils induce angiogenesis through expression of IL-4, IL-5, IL-10 IL-13 and IL-8) [208,209] and in human non-small-cell lung cancer (NSCLC) [156] (Table 1), through of the inhibition of T and NK cell function by IDO [210].

Concerning pro-tumourigenic phenotype of basophils; whereas IL-4 in its anti-tumourigenic phenotype was able to reduce M2 macrophage polarisation [189] in pancreatic ductal adenocarcinoma (PDAC) [187] and in gastric cancer patients [188]; it was also found to promote M2 macrophage polarisation, leading to the formation of an immune-evasive TME. That was observed in lung cancer [211] (mediated by IL-13) or in OSCC cells [212] (mediated by CCL4). More importantly, although CCL3 [157] and CCL4 [191] can contribute to the recruitment of CD8^+^ T cells promoting the anti-tumour immunity, CCL5 [158], in its pro-tumourigenic role, was able to induce immune escape by reducing CD8^+^ T cell infiltration by Treg cell-derived TGF-β in the TME, both in murine and human colon cancer (Table 1). Indeed, another study reported a CCL5 pro-tumour activity in breast cancer pulmonary metastasis [213], associating CCL5 with macrophages recruitment in breast cancer distant colonisation [214]. A recently studied, reported that CCL5 was involved in tumour recurrence through macrophage accumulation in residual tumours, which was in turn responsible for collagen deposition [215].

#### 6.1.2. Mast Cells

▪Anti-tumour immunity

Mast cells (MCs) have been largely overlooked resulting in an actual, still, controversy about its contribution to tumour development and their functions in the TME [98]. Nowadays, is well-known that MCs act as sentinels that monitor the microenvironment in search of stress signals (like DCs and macrophages [216]) and infiltrate different human cancer types either at the peri-tumoural or intra-tumoural level, behaving similar to anti-tumour or pro-tumour cells (or innocent bystander cells) [217] by influencing the tumour biology in a direct or indirect manner. The fact that TAMCs promote or suppress the tumour growth depends mainly on the TME stimuli [217], although the tumour type [100,102,218,219], the tumour stage [220], and the MCs location within tumour tissue are also important factors [93,95].

The stimuli of the TME that contribute to MCs recruitment into the tumour can be, among others, tissue ischemia, hypoxia, cellular injury and soluble factors secreted from the tumour cells and noncancerous stromal cells; however, the molecular mechanisms are still unclear. In this context, tumour-derived factors such as stem cell factor (SCF) (considered the main survival factor for MCs), CXCL12 [221], and CCL15 [222] are involved in MCs recruitment. In addition, several chemokine/receptor interactions such as PGE2 with the Prostaglandin E2 Receptor 2 (EP2 receptor), VEGF via Vascular endothelial growth factor receptor-1 (VEGFR-1), CXCL8/IL-8 interactions with CXCR1 and CXCR2 are of critical importance for the attraction of MCs to the sites of chronic inflammation, including TME [217].

The prognostic value of TAMCs in human solid tumours is still unclear and controversial [98]. Nevertheless, is well-known that MCs through releasing IL-1, IL-4, IL-6, IL-8, TNF-α, IFN-γ, TGF-β, MCP-3, MCP-4, leukotriene B4 (LTB4) and chymase can contribute to inflammation, inhibition of tumour cell growth, and induction of tumour cell apoptosis [223]. Furthermore, prostaglandin D2 (PGD2) secretion is correlated with the inhibition of angiogenesis and vascular permeability in mouse model of lung carcinoma [224]. Currently, it is known that they have been correlated to good prognosis in some human cancers such as NSCLC (independently of tumour stage [225]), breast cancer (independently of tumour grade, age and molecular cancer subtype [100,102]) and prostate cancer [220]. Nevertheless, in this latter, its role depends on MCs location within tumour tissue and tumour stage, correlating intra-tumoural MCs numbers (but not peri-tumoural MCs) with a good prognosis [93,95]. In further support in the context of tumour type-depending TAMCs prognosis, it was reported that in breast cancer patients, peri-tumoural MCs showed cytolytic activity against tumour cells, being related to good prognosis [159] (Table 1).

▪Pro-tumour immunity

When activated as pro-tumoural cells, MCs are able to promote (1) proliferation and survival of tumour cells (either in a direct manner by cell-cell contact or in a direct/indirect manner by releasing of mediators as reported, for instance, in gastric cancer patients involving adrenomedullin and IL-17A) [226]; (2) angiogenesis and lymphagiogenesis by secreting angiogenic components including IL-8, TNF-α, TGF-β, nerve growth factor (NGF), VEGF, fibroblast growth factor (FGF-2) and urokinase-type plasminogen activator (PA) [227]; (3) facilitation of invasion and metastasis by releasing MMP, tryptase (one of the most powerful) and chymase [223]; (4) TME immunosuppression. Interestingly, in this context, intra-tumoural MCs are associated with poor prognosis. They exert pro-tumour activity since they are able to disrupt the anti-tumour immunity in human gastric cancer by expressing a significant higher level of immunosuppressive molecule PD-L1 leading to immunosuppress T cells [160] (Table 1). Hence, MCs have been associated with worse prognosis in CRC patients [219], gastric cancer [228], PDAC cancer [229], in breast [230], lung [231] and prostate cancer patients [232].

#### 6.1.3. Macrophages

▪Anti-tumour immunity

Tumour-associated macrophages (TAMs) are one of the most important components of the tumour immunosuppression microenvironment [233]. Owing to their heterogeneity, plasticity and the integration of the epigenetic memory of these cells, they can polarise into different phenotypes over time, i.e., during tumour progression, the macrophage phenotype changes from classically activated (M1 macrophage) to alternatively activated (M2 macrophage) [234] or even to patterns distinct from those of M1 and M2 macrophages, according to developmental origin, tissue of residence and surrounding microenvironment [235,236,237]. Hence, as in the case of neutrophils, macrophages cannot only be classified into M1 and M2, but the acquired phenotype by polarisation goes beyond a simple dichotomous classification due to their complexity in the real TME, making them a “double-edged sword”.

During initial stages of tumour development, macrophages can either directly promote anti-tumour responses by killing tumour cells or indirectly recruit and activate other immune cells [238]. In the presence of a tumour, these immune cells phagocytose tumour cells and process them into antigen peptides, which will be presented to MHC-II molecules on their surface. Consequently, T-cells are stimulated, increasing their proliferation and activation with the synergistic effect of co-stimulatory molecules [239]. Nevertheless, they do not migrate to the lymph nodes and are unable to activate T cells ex vivo [240]. Macrophages together with DCs, in solid tumours, are the most important cells regarding antigen uptake and presentation [240], being clearly, major factors in driving T-cell expansion and differentiation. For this to take place, it is necessary a cooperative co-stimulation by membrane-bound B7 (co-stimulatory molecule) and macrophages-secreting IL-12 [241]. IL-12 is involved in promoting the link between innate resistance and adaptive immunity by stimulating anti-tumour responses in several models of solid tissue tumours [242,243]. IL-12 also induced the production of IFN-γ, which in turn, play a role in the significant proliferation of Th1 [241,242] and in modulating CTLs and NK cytotoxic activity including the expansion and survival of activated T-cells and NK cells [244]. In addition, a study reported that IL-12 treatment is able to influence TAM function by immunosuppressive environment conversion to tumour regression, including a support for cytotoxic activity (activated NK and CD8^+^ T cells) [243].

M1-polarised macrophages (characterised by high TNF and iNOS expression [238]) mediate both antibody-dependent cellular cytotoxicity (ADCC) and macrophage-mediated cytotoxicity [233]. This latter can take place through several ways: (1) through cell-to-cell contact (it is the most efficient), requiring the participation of anti-tumour antibodies and partly regulated by NO (nitric oxide) [245,246]; (2) due to several soluble factors (anti-tumour products secreted by macrophages) such as nitrite ion [245], TNF, Fas ligand (FasL or CD95L), hydrogen peroxide (H_2_O_2_) [233,245], iNOS (also known as NOS2) or high levels of MHC-II molecules [238].

Regarding factors that can modulate macrophage function, NO is of pivotal importance. It is produced by NO synthases (one inducible enzyme (iNOS or NOS2) and two constitutive enzymes (cNOS or NOS1) [247,248]. NO flux can change depending on the presence of cytokines. Thus, in presence of IFN-γ and lipopolysaccharide (LPS), NO concentration can be high (anti-tumour, anti-angiogenic and anti-pathogen functions). Conversely, the presence of IFN-γ and IL-1β is related to lower NO concentrations (critical in wound healing responses exploited by the tumour during disease progression) [246]. High levels of NO were related to anti-tumoural activity in murine leukemic cells (apoptosis) and human multiform glioblastoma (necrosis) [246]. In addition, iNOS overexpression, which lead to high levels of NO, was reported as anti-tumour immunity in mice thyroid cancer (tumourigenesis inhibition); xenograft mice prostate (inhibition of tumour growth) and xenograft mice pancreatic (apoptosis), among others [161] (Table 1). In addition, other factors such as macrophage colony-stimulating factor (M-CSF) and CCL2 promoted macrophage recruitment and are also implicated in determining macrophage phenotype [238]. Furthermore, it should be pointed out that the expression of Protein kinase N2 (PNK2) is involved in regulating TAMS since it is able to promote M1 macrophage and suppress tumour-associated M2 macrophage polarisation and tumour growth via regulating Dual-specificity phosphatase 6-extracellular signal-regulated protein kinases 1 and 2 (DUSP6-Erk1/2) pathway in human colon cancer tissues, thereby being related to good prognosis [249].

However, the TME, such as nutrient availability, fibrosis, hypoxia, and lymphocyte-derived factors, appear to shift macrophage phenotypes most dramatically [238].

▪Pro-tumour immunity

In contrast, pro-tumourigenic phenotype expressed high levels of IL-10, in addition to high concentrations of ARG1, cluster of differentiation 163 (CD163), CD204 or CD206 [238]. Whereas M1-polarised macrophages contribute to anti-tumour immunity, M2-polarised macrophages are involved in promoting cancer cell growth, angiogenesis, metastasis and establishment of an immunosuppressive environment [239], being thus associated with tumour progression and poor prognosis [250]. Nevertheless, M1 macrophage, in a situation of chronic inflammation (as in the case of neoplasms), can also adopt the pro-tumourigenic phenotype by inducing tumour initiation through a mutagenic microenvironment, and M2 can promote malignancy progression [239]. TAMs generally represent a major component of myeloid cells present in the TME [239]. As early as 1970s, it was found that they play a key role in tumour growth [251], associating their accumulation with poor prognosis in solid tumours [250,252].

TAM infiltration-derived cytokines are able to contribute to the pro-tumoural environment mediating tumour cells proliferation and survival, such as hepatocyte growth factor (HGF), EGF, platelet-derived growth factor (PDGF), epithelial growth ligands of the factor receptor (EGFR), TGF-β1, among others [233]. M2 macrophages, may stimulate tumour growth in vivo [253], since, they are involved in promoting metatastasis by expression of MMPs, serine proteases, cathepsins, and decompose various collagens, and other components of ECM and angiogenesis by coordinated expression of VEGF, basic fibroblast growth factor (BFGF), IL-1,IL-8, TNF-α, MMP2, MMP9 and NO [233]. The release of exosomes by macrophages can also lead to cancer metastasis by transferring certain micro-ribonucleic acid (miRNAs) into cancer cells, such as CRC [254] and PDAC cells [255]. Furthermore, inhibitory targeting of macrophage signaling pathways might be associated with better prognosis of cancer patients. In this context, the Phosphatidylinositol 3-kinase (PI3K) signaling pathway, which is of vital importance in the regulation of cell survival, but also in the malignant transformation of cells, was also involved in the pro-tumoural activity of macrophages, as reported in several cancers such as human head and neck squamous cell carcinoma (HNSCC) [256] and human chronic lymphocytic leukemia (CLL) [257]. PI3Kγ regulates NF-κB and CCAAT/enhancer-binding protein-β (C/EBPβ) during macrophage polarisation, thereby regulating macrophage immune responses. In this regard, macrophage PI3Kγ acts as a switch during immune stimulation and suppression during inflammation and cancer. The PI3K/AKT (protein kinase B) and the mammalian target of rapamycin (mTOR) signaling pathways (PI3Kγ/AKT/mTOR) were able to inhibit NF-κB activation and stimulate C/EBPβ activation, thereby promoting immune suppression during tumour growth. Nonetheless, inactivation of macrophage PI3Kγ resulted in immune stimulation (re-establishment of effector CD8^+^ T cells, among others), implying the presence of NF-κB activation and the absence of C/EBPβ activation [256]. Other pathways and factors also involved in determining the phenotype macrophage are: (1) Very late antigen-4/vascular cell adhesion molecule-1 (VLA4/VCAM1) pathway: macrophage-mediated vascular permeability determines ascites development in human ovarian cancer through this pathway [258]; (2) tumour-derived exosomal miR-934 (microRNA-934): its overexpression is related to human CRC liver metastasis by inducing M2 macrophage polarisation activating PI3K/AKT signaling pathway and the downregulation of phosphatase and tensin homolog (PTEN) expression [259]; (3) receptor for activated C kinase 1 (RACK1): M2/M1 ratio is increased through RACK1 by regulating NF-κB with subsequent tumour progression as reported in OSCC patients [260]; (4) wingless-related integration site (WNT) pathway: the activation of paracrine WNT signaling by macrophages in mice ovarian cancer stem cells (CSCs) stimulated CD206^+^ M2 macrophage activation in the TME, thereby promoting to highly malignant and metastatic disease [261]; (5) Signal transducer and activator of transcription 6 (STAT6) pathway: its suppression inhibits TAM differentiation and, thus, their pro-tumourigenic activities, i.e., tumour growth and metastatic niche formation, as reported in mice breast cancer [262]. In support of this, STAT6 signaling pathway demonstrated to be a useful and potential therapeutic target since Gefitinib was capable of effectively inhibits M2-like polarisation by targeting this pathway in Lewis lung carcinoma in mice [263].

In addition to metastasis, tumour growth and angiogenesis, they can also contribute to tumourigenesis by generation of inflammatory Th subset [233] and induction of therapeutic resistance [239]. Recently, it was also found that TAMs have interactions with cancer stem cells, resulting in tumourigenesis and metastasis [264]. The capacity of TAMs to promote angiogenesis has been observed in human esophageal SCC (squamous cell carcinoma) [265], gliomas [266], bladder [267], breast [268], and prostate cancer [269].

In addition, alveolar macrophages can promote hepatocellular carcinoma lung metastasis in mice (through LTB4-inflammatory mediator) [270] and induce metastasis in mice breast tumour cells (through Th1 responses suppression) [162] (Table 1).

Regarding immunosuppressive environment promoted by M2 macrophages, they can inhibit CD8^+^ T cell proliferation in a direct or indirect manner. Regarding direct manner, they can carry it out through three distinct mechanisms: (1) metabolism of L-arginine via ARG-1, iNOS, ROS/RNS [233] causing Treg/Th17 imbalance mediated by exosomes consisting of various miRNAs derived from TAMs [239]; (2) immune checkpoint engagement via expression of molecules such as PDL-1 [238]; (3) production of inhibitory cytokines such as IL-10 and TGF-β [233]. On the other hand, in an indirect manner, they control the microenvironment by TAMs, which are able to recruit immunosuppressive populations (such as Treg) or inhibit stimulatory populations (such as DCs) [238].

Tumour cell-derived soluble molecules implicated in promoting macrophage pro-tumourigenic phenotype (TAM M2 polarisation) are sonic hedgehog (SHH), involved in suppressing the recruitment of CD8^+^ T cell, expression of osteopotin (OPN) in tumour cells and mucin 1 (MUC1) involved in inducing M2 phenotype [233]. Additionally, chemo attractants present in the TME (IL-4, IL-13, TGF-β, and IL-10) lead to the adoption of an M2 phenotype [264].

The metabolic conditions such as alterations in glucose, lipid and glutamine metabolism can promote M2 macrophage phenotype, even M1 conversion to M2, mediated by lactate secreted by glycolisis in cancer cells [233].

In different types of cancer such as human breast cancer [271], xenograft mice in endometrial cancer [272] and renal cell cancer patients [273] a synergistic relationship has been demonstrated between TAMs and tumour cells which can lead to angiogenesis. Supporting this synergistic relationship, a study found out that the tumour cells produced the necessary stimuli to initiate tumour angiogenesis, but the initiation is delayed if TAMs are not present [264]. All these observations show that the potential cytotoxic activity of macrophages might be modulated by the TME, thus dominating macrophages’ pro-tumour activity [274].

#### 6.1.4. Dendritic Cells

Originally, DCs were classified depending on the anatomical location and physiological conditions [275]; nevertheless, recently, they have been divided into five subsets according to species and anatomical locations. Thus, currently, they can be nominated as follows: conventional DCs 1 or 2 (cDC1s, cDC2s); plasmacytoid DCs (pDCs); monocyte-derived DCs (moDCs) and Langerhans cells (LCs) [276].

▪Anti-tumour immunity

As above-mentioned, DCs are recognised as the bridge linking adaptive and innate immune responses, being at the centre of immune responses as they are the most efficient cross-presenting cell type [277,278]. Accordingly, they are recognised as major players in the control of cancer by adaptive immunity [279]. This is achieved via cross-presentation by MHC-I and MHC-II of tumour-derived antigens to CD8^+^ T cells or CD4^+^ T cells, respectively (Figure 3). To activate T cells, there are several features that are necessary, such as the cross-presentation [280,281,282,283], co-stimulatory signals (PAMPs, DAMPs) and activating cytokines that contribute to maintain and boost T cell function within tumours and expand them [278,284,285,286] (Figure 3). Regarding cytokines, although it was thought to be secreted only by DCs, currently it has been suggested that APC can co-operate with others myeloid cells to carry out this cytokine production (IL-12 and IL-15, among others) [98].

The importance of cross-presentation and cross-priming of specific CD8^+^ T cells in anti-tumour immunity was firstly reported in 2003, inducing apoptosis in tumour cells [287]. In this context, from the 21st century, CD103^+^ cDC1cells are considered to be the main subset cross-presenting antigens from peripheral tissues such as skin, lung and intestine [282]. Recent studies have shown that mice deficient of cDC1 cross-presentation fail to reject highly immunogenic fibrosarcoma tumours [288]. Despite the fact that they are the minority DC subset in tumours [98], cDC1 cells has been associated with good prognosis in different contexts, such as human and mouse lung tumours [289], human hepatocellular carcinoma [163] (Table 1), murine breast tumours [290] and human melanoma tumours [291].

Moreover, a study carried out in mice reported that CD103^+^ cDC1-derived CXCL9/CXCL10, are capable of recruiting effector and memory T cells expressing CXCR3 [98,278,292]. In addition, they are the only population that mediate the tumour antigen trafficking both from TME to tumour draining lymph nodes and from tumour-specific effector T cells into tumour, in the context of solid tumours [278].

On the other hand, the correlation between high CCR7 (expressed on cDC1s) expression level (both human and mice) melanoma and patient survival has been reported, due to its pivotal role in trafficking tumour antigen to lymph nodes and the recruitment of CD8^+^ T cells to the tumour and the proliferation of T cells in TME [293].

DCs also play a key role in the recruitment of other immune cells through activating cytokines [98]. Hence, cDC1s-derived IL-12 within tumours [240,294] induced IFN-γ production by CD8^+^ T cells [295] as well as NK cell-mediated cytotoxicity [292,296,297], as described in human melanoma [291] (Figure 3). Of note, not only NK cells are important for cDC1 anti-tumour activity but also the localisation of leukocytes within the tumour, which are of pivotal importance for cDC1 function [292]. Indeed, in 2020, a study carried out in mice, reported that the reestablishment of IL-12 expression, involving a previous IL-4 blockade, by a new subset of regulatory DCs resulted in a tumour control, due to enhancement CD8^+^ T cell function [298]. It should be emphasised that the function as APC by cDC1s in mouse models has been reported to be more efficiently than in humans [299].

On the other hand, unlike cDC1s, cDC2s are not as potent at cross-presentation; however, are the most abundant population in lymphoid and non-lymphoid tissues [299] and are better activators of human CD4^+^ T cells than cDC1s [292], thereby increasing the priming CD4^+^ T cells [299,300]. Recently, it has been reported that human cDC2 (also cDC1) can negatively induce the functional effects on T cell priming by IDO expression [301].

High cDC2s (along with low Treg infiltration) in the TME of human HNSCC [300], in human melanoma [293] and in breast cancer patients [240] were significantly associated with longer progression-free survival. Importantly, cDC2s are dispensable for CD8^+^ T cell activation and proliferation in some tumours. In addition, in lung and CRC patients, tumours cDC2s might help in eliminating myeloid-derived suppressor cells (MDSCs) and shifting the TAM phenotype from M2-like to M1-like [165]. Recently, cDC2 has been positively associated with inflammasome disorders (condition of various cancers) [302].

Concerning pDCs, they have the capacity to be pro-tumour or anti-tumour depending on their environmental stimuli [98]. Although, the role of pDCs in human tumours is less well established than that of the cDC subsets, several studies have associated them with promoting anti-tumour activity owing to the production of IFN-I [278]. APCs function and cooperation with cDCs, resulting in an optimal cross-priming [303,304]. In this context, it was reported that the absence of IFN-I or the inability of cDC1 failed CD8^+^ T cell response (and thus, impairment in immune rejection of tumours) indicating that IFN-I signals are required for anti-tumour CD8^+^ T cell responses through DCs [305].

Regarding moDCs, it has been suggested that they are of vital importance in sustaining an immune response during certain inflammatory conditions (their recruitment enhances TNF and iNOS-producing inflammatory DCs). Despite the fact that their role in the development of spontaneous anti-tumour immunity is unclear [292], it has been described their tumour immunity activity in lung and CRC patients [165] (Table 1), wherein moDCs may efficiently scavenge tumour antigen, but are unable to migrate to the tumour-draining lymph nodes. The monocyte-derived inflammatory DCs (inf-DCs) contribute to anti-tumour activity by activating CD4^+^ and CD8^+^ T cells [306,307].

Lastly, Langerhans cells were discovered in 1868 by Paul Langerhans but it was only in the 1990s that their function similar to APCs was identified. Moreover, their presence in the TME has been recognised in a range of cancer types, particularly skin cancers, in which their amount is lower compared with normal skin. Its good prognosis has also been reported (among others) in primary lung adenocarcinoma patients (Table 1), wherein can act as APCs (directly to other infiltrating cells or through macrophages to memory lymphocytes) resulting in T cell activation after their migration into regional lymph nodes and in primary invasive ductal breast carcinoma patients [98]. Its anti-tumour role has been also highlighted in stage III gastric cancer, resulting in significant overall survival of the patients [308]. They can be regulated by changes in the cytokine profile and other secreted molecules, as well as by downregulation of surface molecules on cells and hypoxia. Thus, IL-10, TGF-β, IL-1β and VEGF may influence in the recruitment and migration in the TME. Langerhans cells are associated with infiltration of immune cells into the tumour, being capable of recruiting NK and CD8^+^ T cells [98].

▪Pro-tumour immunity

Tumour-infiltrated DCs (TIDCs) can be influenced by the TME [309,310], leading to tumour-infiltrating DCs conversion into immunosuppressive regulatory cells, capable of behaving as immunosuppressive modulators. In this context, they can confer immune suppression instead of immune stimulation at the local TME involving evasion immune surveillance [278]. In support of this, cytokines such as IL-6 and IL-10 can influence in the development and function of DCs. Tumour cells-derived IL-6 production has the ability to disrupt the DC maturation, migration and differentiation [311,312,313]. Furthemore, IL-10, considered as kind of anti-inflammatory interleukin can also be produced by DCs, suppressing the killing ability of immunity to tumour [314]. In this context, it has also been demonstrated that tumour-derived CXCL1 and CXCL5 influence in DC maturation, as reported in human CRC [164] (Table 1).

On the other hand, in solid tumours, IL-10 production by components of the TME (such as TAMs) [315] can, among others, contribute to downregulate CD103^+^cDC1 tumour-infiltrating DCs-derived IL-12 production [316]. In addition, other factors such as VEGF, TGF-β, IL-1β, IL-13, Granulocyte-macrophage colony-stimulating factor (GM-CSF) and prostaglandins produced by tumour cells (and other cells of the TME), impact in DCs differentiation and/or maturation [278]. TGF-β is capable of inducing the conversion of T cells, recruited by DC expressing IDO previously, into Tregs, generating an immunosuppressive microenvironment that favours tumour progression [317,318] (Figure 3).

Regarding pDCs, their ability to produce IFN-I can be disrupted by contact between immunoglobulin-like transcript 7 (ILT7) on pDCs and bone marrow stromal antigen 2 (BST2) on tumour cells. Tumour cells-derived VEGFA, which contribute to promote tumour growth and metastasis, is involved in the inhibition of DCs maturation [98]. If pDCs adopt the pro-tumourigenic phenotype, they can contribute to tumour growth, invasion, and/or angiogenesis stimulation by promoting the differentiation of naïve CD4^+^ T cells into T cells secreting IL-10, involving immunosuppressive functions [305,319,320], among others.

As pro-tumourigenic cells, they can induce the generation of Tregs in the TME and tumour-draining lymph nodes [321] as described in breast cancer progression [322]. In this context, pDCs have also been associated with poor prognosis in ovarian carcinoma patients by inducing IL-10-producing CD8^+^ Treg cells and inhibition of T cell proliferation [292], resulting in an induction of angiogenesis through production of TNF and IL-8, as reported in human ovarian cancer [323].

Concerning moDCs, they can loss its anti-tumour activity and may contribute to an immunosuppressive environment at the tumour site in lung and CRC patients [165] and in human SCC microenvironment (Table 1) wherein TGF-β, IL-10 and VEGF-A can suppress DC function resulting in impairment of T cell activation [166]. In addition, higher levels of CD14^+^ and BDCA1^+^ CD14^+^ cells (moDC) have been found in melanoma skin metastatic lesions in comparison with healthy skin [324].

Regarding Langerhans cells, their pro-tumourigenic phenotype can support epithelial-mesenchymal transition (EMT) in cutaneous cancers involving TGF-β [325]. Hypoxia can also influence LCs functions, in the context of stimulating T-cell responses, for instance, leading to tumour cell evasion [98]. In relation to FoxP3^+^ Tregs recruitment, they do not have a direct role, but they may express a tolerogenic role in human melanomas [326]. Whereas IL-10 inhibit LCs migration [98], IL-1β contribute to their proliferation and their pro-tumourigenic cytokine network is stimulated [167] leading to tumour pathogenesis [98] through the dysregulation of signalling pathways in human dysplastic oral keratinocytes (DOK), as demonstrated in OSCC cells [167] (Table 1). In general, LCs contribute to carcinogenesis and angiogenesis, involving 2,4-dimethoxybenzaldehyde (DMBA) and pericyte-derived milk fat globulin E8 (MFG-E8), respectively. Nevertheless, their role in lymphangiogenesis has not been defined, although it is known that they contribute to lymphatic vessel formation in skin [98].

#### 6.1.5. Innate Lymphoid Cells (NK Cells and Helper ILC1, ILC2 and ILC3)

Recently, in 2018, the ILCs cells have been classified into five subsets, according to their function and transcription factor requirements. ILCs are represented by a cytotoxic activity mediated by NK cells and by a helper activity mediated by ILC1, ILC2, and ILC3 cells, representing the innate counterpart of T lymphocytes. In addition, lymphoid tissue inducer cells (LTis) are other subset of ILCs of pivotal importance in the formation of lymph nodes and Peyer’s patches during embryonic development, carried out, mainly by lymphotoxin. ILC Group 1 consists of NK cells and ILC1s, Group 2 consists of ILC2s and lastly, Group 3 comprises ILC3s and LTis [174,327].

##### ILC Group 1 (NK and ILC1s Cells)

Natural Killer cells

▪Anti-tumour immunity

NK cells can act spontaneously, that is, they can execute their cytotoxicity function without the need for neither prior sensitisation (pre-activation) [98]. In addition, their ability to produce IFN-γ and TNF-α rapidly upon cell triggering, results in an early response to oncogenic transformation involving the death of appropriate cellular targets. Hence, its prompt intervention against tumour cells, including their role to favour the initiation of inflammation, is associated with their active participation in the immunosurveillance of tumours [297,328]. Nevertheless, in order for NK cells to carry out their cytotoxic function optimally in the TME, they must first have been triggered into an activated state by contact with the ECM [329].

NK cells are implicated in oncolysis through their cytotoxic response which is divided into four major steps: (1) recognition of the target through the immunological synapse between NK cells and the aforementioned target, resulting in the reorganisation of the actin cytoskeleton; (2) lytic synapse that involve the polarisation of microtubule organising centre (MTOC) and secretory lysosome; (3) Docking process (not complete, that is, not total fusion) between secretory lysosome and the plasma membrane of NK cells; (4) Complete docking process (total fusion) releasing their cytotoxic contents (such as perforin and granzyme) towards the target cell plasma membrane in a process called degranulation. Regarding the aforementioned cytotoxic contents, perforin-dependent cytotoxicity is crucial for NK cell-mediated control of several tumours [330]. In fact, this was evidenced when the role of perforin-dependent NK cell activity in the control of B cell lymphomas and mice mammary carcinoma was studied [331]. Additionally, NK cells display memory features and inhibit tumour growth in IFN-γ and perforin-dependent manner [330].

In addition to the release cytotoxic granules, death receptor-inducing target cell apoptosis is another anti-tumoural method used by NK cells (involving TNF-related apoptosis-inducing ligand (TRAIL)/FasL expressed by tumour cells) [332].

It should be emphasised that NK receptor act in an MHC-dependent way (MHC-restricted), recognising virus-infected cells or tumour cells by activating receptors (killer cell imuunoglobulin-like receptors) (KIRs) in humans and Ly49 receptor family in mice due to the loss of MHC-I as an immune-escape mechanism, not expressed in healthy cells on their surface [98,333].

NK cells have the ability to control tumour growth, as demonstrated in human specimens in the 1980s [334,335], either directly (direct contact between NK cell and tumour cell) or indirectly (affecting the function of other populations of innate and adaptive immunity in the TME). To support this notion, NK anti-tumour activity in leukaemia mouse model was reported, in the context of elimination of tumour cells by NK cells [168] (Table 1) and increased cancer metastasis and also lower resistance to infection in patients with inefficient NK cells [336]. Later, in 2016, a study showed that NK memory cells were able to control acute myeloid leukaemia (AML) growth in mice [337]. Furthermore, it was reported that in CRC, NK cells can target and eliminate CSCs, a subset of cells with self-renewal ability involved in the generation and evolution of tumours, thus controlling tumour development and dissemination [338,339].

Effector immune response of NK cells involves the secretion of a wide variety of cytokines (IFN-γ, IL-10, IL-5, IL-13, TNF-α, and GM-CSF) and chemokines (macrophage inflammatory protein-1 alpha (MIP-1α), MIP-1β, IL-8, and CCL5). It is worth noting that molecules such as CD96, CD161, and CD244 expressed by NK cells can act as co-receptors and regulate cytolytic activity and cytokine production, being IFN-γ as one of the most frequent cytokines that are secreted by NK as result of activating receptor NKG2D (Natural Killer Group 2D) on their surface, that play a pivotal role in anti-tumour activity, due to its anti-proliferative, anti-angiogenic, and pro-apoptotic effect [330].

On the other hand, factors that regulate NK cell function are TME-derived cytokines (IL-2, IL-12, IL-15, IL-18 and IFN-I) and DAMPs. Among the TME-derived cytokines (secreted mainly by myeloid cells and T cells), IL-12 and IL-18 can promote NK cell function when acting in a synergistic manner; IL-15 and IL-2 are both implicated in inducing NK cell cytotoxic function and IL-12, IL-15 and IL-18 if combined can induce “memory-like” NK cells [340].

Importantly, NK and DCs can co-operate within the TME. This is because NK-derived release of apoptotic bodies and DAMPs after death of tumour cells, promote DC function regarding antigen uptake and T cell priming, in addition to interact with DCs directly to induce protective T cell responses and eliminate immature tolerogenic DCs. Hence, NK cell-derived cytokines and chemokines can have an impact on both the innate and adaptive immune responses [98].

During the 21st century, several studies have shown that high levels of tumour-infiltrating NK cells has been associated as an indicator of good prognosis in breast cancer [341], neuroblastoma [342], gastrointestinal stromal tumour [343], prostate cancer [344] and HNSCC patients [345]. Nonetheless, different factors contribute to attenuate the anti-tumoural properties of NK cells, being the most important the tumour type and stage [346]. In this context, higher infiltrates of NK cells were found in lung metastases of renal cell carcinoma than for CRC [347]. In addition, in endometrial cancer, intra-tumoural NK cells were found in 60% of the patients, whereas they were not detected in the remaining 40% [348].

Additionally, it has been demonstrated that although, in lung tumours gluconeogenesis enzyme fructose bisphosphatase 1 (FBP1) can lead to dysfunctional NK cells, pharmacological inhibition can reverse to functional NK. Indeed, it was found that NK cell function depends on stage tumour, thus, NK cells are functional during tumour initiation, mild dysfunctional during tumour promotion and irreversibly dysfunctional subsequently, being impossible to rescued them by FBP1 inhibition [349]. Importantly, tumour progression not only influenced in NK cell function but also in NK density, as reported in studies carried out in early stage of mouse melanoma where a significant higher number of NK cells in the TME compared with primary melanoma was found [330]. In addition, NK anti-tumour immunity may also depend on the concurrent expression of ligands for NK cell receptors in the TME, as reported in endometrial cancer [348].

▪Pro-tumour immunity

Despite the well-known contribution of NK cells in anti-tumour activity, due to the influence of TME in an attempt to evade NK cell surveillance, they can be regulated and adopt the pro-tumourigenic phenotype. The mechanisms implicated in their role as pro-tumourigenic cell are related to immunoediting and immunosuppression [98].

On the one hand, in support of the influence of TME in NK receptors, a study showed that the neutralisation of the TME restored the cytotoxic activity of NK cells enhancing NKG2D expression [349], being the NK cells immune checkpoint receptors, the source of immune escape for various cancers [350]. On the other hand, regarding the immunosuppression, the TME induced an immunosuppressive environment through inefficient or tolerogenic immune cells, as in the case of Treg, MDSCs, TAMs and through soluble factors such as PGE2, TGF-β, IL-6, IL-10, IDO, dickkopf-related protein 2 (DKK2), high concentration of adenosine, large amount of lactate [349], NO, exosomes, secreted miRNAs [91], soluble human leukocyte antigen G (HLA-G), soluble NKG2D ligands, and galactin-3 (soluble inhibitory receptor for NKp30), which are an obstacle to proper NK function [98,350].

In this context, the cytokines secreted by other immune cells or stromal cells in the TME can influence the anti-tumour function of NK cells not only positively but also negatively, leading to NK cell behave in a pro-tumoural manner, as reported in CRC patients [330,349]. In addition, the TME of solid tumours escape from NK function by preventing the recruitment of intra-tumoural NK cells and allowing only the recruitment of non-cytotoxic NK cells in the TME or in the case of intra-tumoural effectors, NK cells could be confined in the stromal part of the tissue [349].

Although NK cells can recognise the tumour cells by loss of MHC-I, in situations of hypoxia, MHC-I can shed from the surface of cancer cells resulting in a downregulated expression of NKG2D and CXCR, serving as a mechanism of resistance to NK-mediated killing [91]. Indeed, as reported in gastric cancer [91], in situations of hypoxia, TGF-β is upregulated, being capable of targeting mTOR, which is required for NK cell function, particularly IFN-γ production [349]. Interestingly, TGF-β was able to transdifferentiate NK cells into ILC1, which is unable of developing cytotoxic functions [349].

Concerning PGE2, although recent [351,352] and past [353,354] studies showed that this soluble factor is able to suppress the cytolytic functions of NK cells, it is still unclear the direct interaction between PGE2 and NK activating receptors. However, in 2010 [355], the capacity of PGE2 to inhibit NK activating receptors was reported. In addition, PGE2 is capable of disrupting the recruitment of cDC1 into the TME by NK cells [297]. Interestingly, the capacity of NK function to be restored by PGE2 blockade was reported in a murine model of metastatic breast cancer [356] and in human gastric cancer cells [169] (Table 1).

ILC1s

▪Anti-tumour immunity

It should be emphasised that although the role of NK cells in controlling tumour growth and metastasis is currently well established, the involvement of helper ILCs in tumours remains poorly understood [174]. Likewise, NK cells, the others ILC1s are dependent on the transcription factor T-bet and potent producers of pro-inflammatory cytokines (e.g., IFN-γ (as signature cytokine) and TNF-α), allowing them to contribute to immune protection [357]; however, differs from NK by their ontogenic paths as well as their cytotoxic machinery [327]. Of all subsets of ILCs, ILC1s have the highest plasticity, thus, they are capable of changing their functional capacity, depending on soluble factors of the TME [174]. ILC1s mainly exert anti-tumour activity, indeed, ILC1-derived IFN-γ can contribute indirectly to anti-tumoural immunity by recruiting and activating effector immune cells through the cytokine production, upregulation of co-stimulatory molecules and cytotoxicity [170,174]. Recently, a review showed that ILC1s-derived IFN-γ can show a protective role against carcinogenesis in the early stage of human multiple myeloma [170] (Table 1).

▪Pro-tumour immunity

ILC1s in their pro-tumoural phenotype contribute to chronic inflammation. Its signature plasticity has been demonstrated in both, in murine and in human studies. In human decidua, molecules present in the TME, such as TGF-β can induce NK cell conversion to ILC1s with pro-angiogenic and immune tolerant features [174], as described in mouse melanoma. In human and mouse NSCLC (eomesdormin downregulation in NKp46 + NK1.1 + of ILC1s) and CRC patients (because of inhibitory receptors expression, such as Klre1 and Klra7) ILC1s have also associated with pro-tumoural functions [171] (Table 1).

##### ILC Group 2 (ILC2s)

▪Anti-tumour immunity

ILC2, ILC3 and LTis are pro-tumourigenic in nature [170]. ILC2s are characterised by producing type-2 cytokines (IL-4, IL-5 and IL-13) [174] being involved in allergy, asthma or parasitic infections [358]. Although, type 2 responses by ILC2s have classically been associated with the promotion of TME; in a study carried out in a mouse model of lymphoma, it was reported that if ILCs were activated by stimuli such as IL-33, they could develop an immune protection resulting in decreased tumour growth [359] and in an indirect support of anti-tumour immunity by enhancing both, MHC-I expression on DCs as well as T cell cytotoxicity, as described in primary and metastatic murine lung [172] (Table 1). In support of this, in mouse lung cancer [171], also IL-33 was related to good prognosis since it could elevate the frequency of tumour infiltrating ILC2s via CCL5, CXCL10, and CXCL12. In addition, also IL-5 has been correlated with anti-tumour activity since is capable of suppressing melanoma growth in the lung by recruiting eosinophils [360].

▪Pro-tumour immunity

ILC2s pro-tumour activity has been shown in human bladder cancer [173] (Table 1) and in acute promyelociytic leukaemia [361] since have been associated with MDSCs immunosuppressive function. Moreover, unlike IL-5, IL-13-producing ILC2s had been associated to poor prognosis in breast cancer through immune suppression limiting anti-tumour T cell responses. In addition to that mechanism, ILC2s can also enhance T-reg functions by amphiregulin (AREG), an EGF-like growth factor [327].

##### ILC Group 3 (ILC3s and Lymphoid-Tissue Inducer Cells)

ILC3s

▪Anti-tumour immunity

ILC3s are dependent on RORγt (RAR-related orphan nuclear receptor γt) for their development and function [174]. They can regulate homeostasis or adopt disease-driving roles in a range of inflammatory disorders. This way, when activated by stimuli such as IL-23, IL-1β (among others), are able to produce IL-17, IL-22, IL-8 and GM-CSF, involved in disease-driving roles [174,327]. They can be divided into two subsets according to the expression of the natural cytotoxicity receptor (NCR) NKp46 in mice and Nkp44 in humans, (NCR + ILC3s and NCR−ILC3s, respectively) [327]. NKp46 has been associated with good prognosis in human NSCLC (owing to the formation or maintenance of tertiary lymphoid structures (TLS), release of IL-22, TNF-α, IL-8, and IL-2 and recognition of lung tumour cells and tumour-associated fibroblasts via the NKp44 receptor [174] (Table 1). Moreover, in the B16 melanoma model also has shown good prognosis by IL-12 production associated to tumour rejection [362,363].

▪Pro-tumour immunity

ILC3s pro-tumoural role have been demonstrated in human CRC as IL-17, IL-23 and IL-22 are able to contribute to metastasis involving angiogenic factors (Table 1). In addition to metastasis, IL-22 has been also associated to chemoresistance and tumour growth [174]. Additionally, whereas originally IL-17 and IL-22 regulate homeostasis, they have been associated with colon cancer progression [364]. Moreover, in lung endothelial cells, ILC3s producing IL-17 contributed to metastasis through VEGF, TGF-β or IL-8 expression [327]. Additionally, IL-17 and IL-22 have been associated with tumourigenesis in inflamed colon [365].

Lymphoid-Tissue Inducer Cells

▪Anti-tumour immunity

LTis, with a similar phenotype to NCR−ILC3s, are capable of producing IL-17 and IL-22 to participate in host defense [327]. This subset can promote an effective immune response [366] in an indirect manner by instructing stromal cells (mesenchymal stem cells (MSCs)) through the production of chemokines CCL19, CCL21 or CXCL13 which are in charge of recruiting lymphocytes, thereby modulating blood vasculature and lymphatic vascular system [327]. Additionally, LTis are vital for the formation of lymph nodes, regulated by the release of lymphotoxins [171].

▪Pro-tumour immunity

The pro-tumoural function of LTis have been highlighted in melanoma and breast cancer, since they contribute to immune tolerance to tumours (immune escape) expressing CCL21 [367]. The correlation between MSCs and LTis can also adopt a pro-tumourigenic phenotype since it has been suggested that a high quantity of LTis along with MSCs promote lymph node metastasis in breast cancer [368]; however, this correlation is still unclear because of LTis could act independently of MSCs [327].

LTis are also involved in TLS formation, notwithstanding, a chronic inflammatory state is sufficient to induce TLS formation even in the absence of lymphoid tissue inducer (LTi) cells. Although, tumour-associated TLSs have been correlated to favourable prognosis in the most type of cancers [369], in the case of mice melanoma tumours [367], the formation of TLS by LTis has been correlated to poor prognosis since it carries the recruitment of immune suppressive cells such as Treg cells and MDSCs to suppress anti-tumour immune response.

### 6.2. Adaptive Immune Response

#### 6.2.1. B Cells—Humoral Immunity

▪Anti-tumour immunity

Unlike T cells, until recently, the role of B cells in cancer has been underestimated, and it was only due to the fact that most patients are resistant to immune checkpoint inhibitor (ICI) therapies targeting T cells that their role in cancer became of interest [370], mostly only during the past decade [371].

Germinal centre B cell (GC B cell) differentiation (into memory B cells and plasma cells (PCs) proliferation, activation, and antibody production depend on T follicular helper cells (Tfh) cells [370,372,373]. Similarly, Tfh depends on B cells in most contexts including its differentiation, thus creating a feedback loop, crucial for a proper regulation of humoral immunity [373].

Since the 1970s [374], it was believed that B cells only had a pro-tumour function, since the behaviour of their antibodies, in some murine models, could favour cancer occurrence and spread. Nevertheless, in recent years, further studies have shown its role in anti-tumour immunity [370]. Indeed, in an analysis of 54 cohorts where 25 types of cancer were studied, 50% of the studies were reported positive with respect to prognostic impact of tumour-infiltrating B cells (TIL B cells). The remaining percentages were 9% deleterious and 41% neutral [375]. In metastatic melanoma, TIL B cells were considered second-best predictor of positive disease outcome after CD8^+^ TILs (tumour infiltrating lymphocytes). Moreover, TILs B cells are also capable of exhibiting antigen-driven clonal expansion, class switching and affinity maturation in the cancer environment [376].

Of note, depending on TME composition, the phenotypes of B cells are present, and their antibodies can behave as anti-tumour or pro-tumour cell [371] (Table 2). TME-derived factors such as CXCL12 and its cognate receptor, CXCR4, are involved in the migration and adhesion of malignant and normal B cells to the bone marrow and secondary lymphoid tissues [377]. As a matter of fact, in recent years, B cells density in tumour, especially those found in the TLS, has been associated with favourable prognosis, being considered an important predictor of outcome for ICIs, even in tumours with low tumour mutational burden [370,378]. In this context, the amount of TLS in the tumour has been directly correlated to positive outcomes in both, human disease and mouse models [379]. Supporting these observations, both in colon cancer [380] and in lung cancer [381], TLS play a role in maintaining an efficient immune response, being the presence of B cells within them a positive predictor of outcome.

B cells produced cytokines and chemokines such as lymphotoxin, (involved in the formation of TLS) [379], CXCL10 [402], CCL4 [403], IFN-γ (increase via TNF-α) [404] and murine ABCD-1 chemokine [405]. A study shed more light on anti-tumour B cell activity, demonstrating that these cells are able to help CD8^+^ T cells through CD27/CD20 [406]. Additionally, the Cancer Genome Atlas (TCGA) database reported the positive correlation between high levels of expression of B cell and PC signature genes and better outcomes, since their presence in patients with PDAC, lung adenocarcinoma, HNSCC and melanoma showed an increase in overall survival [407].

B cells are found in smaller proportions in tumours (25% of all cells in some tumours) compared with T cells; nonetheless, their role in tumours is important to predict therapeutic response in cancer [376]. PCs are also present in tumour infiltrates and despite the fact that they are found in small quantities, they can produce large amounts of cytokines and antibodies [371].

TIL B cells and intra-tumoural produced antibodies can promote anti-tumour activity via the following mechanisms: (1) presenting tumour-derived antigens to CD4^+^ and CD8^+^ T cells [371]; (2) altering the function of their antigenic targets on cancer cells; (3) facilitating opsonisation of tumour cells [376] leading to uptake antigen via Fcγ receptors on the DCs and enhancement of the presentation by DCs [371,376]; (4) activating the complement cascade; (5) via ADCC, resulting in macrophages and NK cell activation; thus, they can contribute to NK cell mediated tumour killing [370]; (6) via ADCP, that is antibody-dependent cellular phagocytosis [370,371]; (7) producing IFN-γ and IL-12 with cytotoxic immune response; (8) by killing tumour cells via granzyme B and TRAIL (also known as TNFSF10) [371].

Regarding effector B cells-derived antibodies (often IgG) [371]; they have demonstrated anti-tumour activity in serum from patients with cancer, as reported in medullary breast cancer (antibodies directed against aberrantly exposed β-actin in apoptotic tumour cells) [376]; lung cancer (anti-p53 antibodies) [376] and ovarian, pancreatic, gastric, breast and lung cancer (antibodies against MUC1, present in serum from patients with cancer at early stages) [371] (Table 2). In addition, in tumour-developing mice (multistage epithelial carcinogenesis), activated B cells were able to promote the innate cells migration into the preneoplastic and neoplastic TME [370].

It is interesting to notice that these tumour specific antibodies have been proposed for early detection of cancer [408,409] and as potent prognostic markers at later stages of disease [410,411]. Furthermore, as mentioned previously, they are considered APCs, thus, this function is also involved in anti-tumour immunity by presenting antigens to CD4^+^ and CD8^+^ T cells and developing an immune response within the tumour microenvironment [412]. PCs are involved in increasing antigen-presentation to T-cells via the IgG antibodies production [370].

Memory B cells could be involved in antigen-presentation as well. Memory B cells circulating in blood have a long-live and serve as support in the control of potential metastatic cells [370]. They also could promote both, T cell expansion as well as memory formation by presenting antigens [413]. Additionally, both effector and memory B cells can perform direct anti-tumour functions via granzyme B and TRAIL, as demonstrated in human hepatocellular carcinoma [371]. Hence, in a primary melanoma was reported that memory B cell score and increased clonality of the BCR repertoire were associated with longer overall survival [414]. Memory B cells can also transfer antigen directly to DCs or via complement receptor 2 to follicular DCs (FDCs) in the form of opsonised immunocomplexes [370]. In addition to the function as APCs, they are capable of forming tumour-associated TLS in the tumour-adjacent TME, which may actively modulate anti-tumour immune activity [378], supporting the development of tumour-specific B and T cells [371].

In general terms, the anti-tumoural capacity of B cells has been described in some types of human cancers such as: CRC (due to the infiltration of CD20^+^ B cells, CD8^+^ T cells and B cell–attracting CXCL13) [415], melanoma (TLS densities) [378], ovarian cancer (CD20^+^ B cells and CD8^+^ T cells immune infiltrates) [416], breast cancer (CD20^+^ infiltration) [417], hepatocellular carcinoma (density of tumour-infiltrating CD19^+^ B cells) [418], HNSCC (peri-tumoural B cell infiltration) [419], or CRC (high infiltration of CD20^+^ B cells increased the prognostic effect of CD8^+^ T cells) [420].

▪Pro-tumour immunity

B cells can induce pro-tumoural activities by complement activation via IL-10 production [370]. One of the mechanisms that contribute to cancer growth and progression is the generation of immune complexes in cancer, as reported in human OSCC cancer [370] (Table 2) which can be generated by anti-tumour antibodies. Although they were originally associated with autoimmune diseases, tumour cell-bound antibodies have been related to cancer [376]. Supporting this fact, a recent study reported that immune complexes led to chronic inflammation (endothelial cells activation) in modality therapy and in tumour cell lysate (TCL) from lung cancer in murine models, resulting in pro-tumour effects by complement activation and increment of angiogenesis via VEGF production by activated macrophages thereby promoting tumour growth and spread. In addition, macrophages can also be activated and to promote pro-tumour activity via pro-inflammatory mediators upon their binding with immune complexes previously generated by B cells [370].

Other studies regarding B cell-derived antibodies reported that B cell depletion with anti-IgM antibodies in murine models of fibrosarcoma and breast cancer, provided a control of metastases [370].

It has been suggested that anti-CD20 used to deplete B cells, can induce to tumour escape since Regulatory B (Bregs) cells evoked by the tumour often express CD20 [421]. Concerning their cytokine production, B cell-derived immunosuppressive cytokines, can inhibit CD8^+^CTLs and NK cells function [370,376]. These cytokines can be: IL-10 (involved along with TGF-β), IL-35 (promote macrophage conversion to pro-tumoural M2 phenotype), IL-21 and TNF-β [376]. Breg cells-derived IL-10 can have a direct negative impact in T cell anti-tumour responses [422] by disrupting the Th1/Th2 balance and inducing DCs to produce IL-4 and downregulate IL-12 and thus IFN-γ [423] (Table 2). IL-12 induce IFN-γ production in T cells and/or NK cells [423], giving rise to attenuate NK cell-mediated ADCC, as reported in patients with multiple myeloma [424]. Bregs-producing TGF-β are able to upregulate ROS and NO production, being crucial to achieve an efficient activity of MDSCs [425]. Interestingly, although in certain tumour settings lymphotoxin can promote anti-tumour immunity, it can be also correlated with pro-tumour activity, since it acts as a survival factor to tumours by promoting tumourigenesis [426].

Breg cells can also exhibit a pro-tumour activity through the lack of response to CTLA4, resulting in a shorter overall survival [414]. Additionally, Breg cells are also capable of producing adenosine, involved in suppression of T cell activation and/or anergy [427]. Higher levels of Breg cells showed, in recent studies, poor prognosis (shorter overall survival) in urothelial urinary bladder and gastric cancer patients. In addition, activated Breg and Treg at the same time correlated with shorter metastasis-free survival in breast cancer [370] (Table 2).

Hence, although Breg cells are a newly designated subset of B cells is well-known that Breg cells are involved in the suppression of anti-tumour immune responses by effector T cell inhibition and the enhancement of Treg cells and myeloid-derived suppressor cells [428].

#### 6.2.2. T Cells—Cellular Immunity

##### CD4^+^ T Cells (T Helper Cells)

▪Anti-tumour immunity

Th cells as central coordinators of the immune response, play a crucial role in the amplification and regulation of the cellular immune response, thus, play an important regulatory role in controlling the initiation and downregulation of the immune response [429]. Naïve conventional Th cell (Th0) is activated via its TCR (which is on its surface) upon receiving and recognising the processed tumour cell antigens through MHC-II of APCs together with cues from the cytokine milieu cells leading to its clonal expansion [430].

Regarding tumours, Th cells have multifaceted roles: (1) provide help for B, NK cells and CD8^+^ CTLs (by secreting cytokines such as IFN-γ and TNF-α); (2) support both humoral and cytotoxic responses and (3) direct anti-tumour activities [245,430]. Due to these features, their fundamental role in developing and sustaining effective anti-tumour immunity, mediating the adaptive immunological response towards cancer and being key regulators in the tumour immune microenvironment cannot be discussed [431]. Additionally, they are also a target for immunotherapies, focusing on their ability to activate a CD8^+^ CTL response [430].

Th1 cells, characterised by the production of IFN-γ, IL-2 and TNF-α, are notable for its capacity to activate macrophages, neutrophils, NK cells and CD8^+^ T cells and are considered, along with the CD8^+^ T cells, the main players against tumours [431], due to their cytotoxic and phagocytic activity, among others [216]. However, it should be noted that, some studies differ on the cytokine production: (IFN-γ and IL-2 [245,432] and (IFN-γ, TNF-α) [430]. Additionally, they are involved in establishing an inflammatory microenvironment [216], necessary to activate immune cells to execute their appropriate functions. Their contribution to anti-tumour activity has been described in cancers such as primary murine melanoma [433] and melanoma patients [382] (Table 2). In addition, as for macrophages, IL-12 stimulates Th1-dominant immunity in vivo [432].

The importance of IFN-γ-producing Th1 cells was suggested since it was observed that they might play an important role in tumour rejection [432]. In accordance with this, previously, it was reported the significance of the link among the following components: IL-12/IFN-γ/Th1, in cancer (and viral diseases) owing to the potency of IL-12 to induce IFN-γ in vivo, which in turn, enhance Th1 function [434].

According to their role in immunotherapies, Th1-polarised CD4^+^ T cells by vaccination with peptide epitopes, and hence, activated CD8^+^ CTLs by boosting CD4^+^ T cell-derived secretion of Th1-characteristic tumouricidal cytokine were used in several studies. In fact, emerging evidence demonstrated the clinical success of CD8^+^ T cell-based immunotherapies [430].

Th2 cells produce IL-4, IL-5, IL-10 and IL-13 [430,431]. As for Th1 cells, also Th2 cells differ in cytokine production in some studies, including either IL-10 [245,430,431,432] or IL-13 [430,431]. These cells are specialised in the activation of eosinophils, basophils and B cells [431]. Although has traditionally been considered as pro-tumour cells [435], recently it has been demonstrated its anti-tumour function, as described in primary human prostate (IL-10 production is involved in angiogenesis inhibition) [384] (Table 2), OSCC [436] and CRC [437] both by eosinophils infiltration). Th2 cells and type 2-immunity take part in tumour immune surveillance and in addition can initiate anti-tumour responses through the MHC-II complex pathway [438].

It was described that Th1/Th2 balance is critically important in various immune responses, including anti-tumour immune responses [439]. The role of Th1 and Th2 cells in anti-tumour immunity was studied, providing that although Th1 and Th2-derived cytokines were equally important with respect to their effectiveness for eliciting complete tumour regression in vivo, further studies demonstrated that Th1 was more important than Th2. This way, characteristics such as the capacity to induce a strong immunological memory suitable for the generation of CTLs, migrate into local tumour sites, produce IFN-γ, and facilitate the induction of anti-tumour CD8^+^ CTLs in vivo suggested their critical role in cancer immune surveillance compared with Th2 cells [432]. Thus, currently, Th1 are considered the main players against tumours, together with CD8^+^ T cells.

Th17 cells, first identified in 2005, are important mediators of autoimmunity and barrier immunity maintenance at mucosal surfaces [440]. Th17-derived cytokines (IL-17A, IL-17F, IL-22, IL-21) are involved in the mobilisation of neutrophils [441]. Tumour growth was enhanced in IL-17^−^/^−^ mice (B16 melanoma and MC38 colon cancer cell lines) [442,443]. In this regard, their functions supporting anti-tumour immunity has been showed in human CRC samples (apoptosis) [388] (Table 2) and murine B16 melanoma cells (inhibition of tumour development) [444]. Moreover, Th17 cell infiltration in some human cancers such as OSCC [445], salivary gland tumour [446] ovarian cancer patients [447], prostate cancer patients [448], lung carcinoma [449], were also related to good prognosis. Moreover, in murine melanoma models, Th17 give rise to Th1-like effector cells resulting in tumouricidal cytotoxicity [450].

Unlike NK cells, these cells do not have the ability to kill tumour cells directly due to the absence of Granzyme B or perforin secretion; nonetheless, they can be involved in cancer immunosurveillance in several indirect ways: (1) by promoting effector CD8^+^ T cells through IL17-A and IL-2 secretion; (2) via the trans-differentiation into T-bet/RORγt and IFN-γ/IL-17; (3) by inducing inflammation both through tumour, stromal and epithelial cells derived-cytokines/chemokines (IL-1β, IL-6, TNF-α, CXCL9/10 and CCL2/20) and (4) by recruiting DCs, NK cells, macrophages and neutrophils leading to supress tumour progression [386]. Furthermore, IL-17F and IL-22 also recruited neutrophils [451]. In addition, regarding metabolic mosdulators, cholesterol anabolism is positively related to their differentiation and agonists for RORγt (transcription factor essential of Th17) may favour Th17-cell-based anti-tumour immune responses. Interestingly, Th17 cells anti-tumour efficacy often depends on secretion of IFN-γ, the signature cytokine of effector T cells [386].

Another subset of cell, Th9, contributes to IL-9 and IL-21 production [430,452]. Previous studies considered IL-9 as signature cytokine of T cells (Th2, Th17, Treg, NK), ILCS and mast cells only; however, nowadays it is well-known that mainly characterises Th9 cells [453]. Recently, has been reported that IL-1β enhances IL-9 production under Th9-skewing conditions [454]. Th9 cells are involved in suppressing solid tumours development via activation (mast cells and NK cells) and recruitment (DCs and CTLS) of several immune cells, promoting DC survival via IL-3 production and killing tumour cells directly via granzyme B and C. They generally exhibit potent cytolytic activity and are even more resistant to exhaustion than Th1. Its anti-tumour activity has been also described in human lung cancer via granzyme [386] (Table 2).

T follicular helper cell (Tfh) a recently identified subset of Th cells [372] mainly produces IL-6 and IL-21 [430] and in addition, can show a cytokine profile such as Th1, 2, 17 and Treg cells [455]. They are involved in long-term humoral immunity [431] i.e., its main role is helping B cells by promoting B cell activation, maturation, and antigen-specific antibody production [456]. Unlike Th1, 2 and 17 cells, Tfh can be reprogrammed to Tfh-like cells [455]. Regarding their functions against cancers, it was reported that in breast cancer patients, Tfh cells have the capacity to organise TLS [390], (Table 2) and in CRC patients, were considered as critical mediators in the immune-pathogenesis of the disease, showing, thus, protective roles leading to good prognosis [457]. In addition to Tfh, a newly defined population of FoxP3^+^ CXCR5^+^ CD4^+^ T cells: Follicular regulatory T cells (Tfr) have been reported recently. Both Tfh and Tfr are implicated in tumour progression in many human tumour sites [456]. Nonetheless, for instance, whereas Tfh was related to a poor prognostic in NSCLC, Tfr reported a good prognosis since it showed an essential immune-suppressive power in the development of NSCLC; although this ability changed depending on tumour stage [458].

Supporting all these reports about Th cells, several studies carried out in melanoma [459] and CRC [460] in the last 5 years have demonstrated that these cells are capable of driving and sustaining meaningful anti-tumour immune responses, as well as supporting anti-tumour CD8^+^ T cell responses. In accordance with this, a Japanese study found a positive correlation between high level of circulating CD62Llo CD4^+^ T cells prior to programmed cell death protein 1 (PD-1) checkpoint blockade and the presence of effector CD8^+^ T cells resulting in a good prognosis. Thereby, it was shown the versatile role of polyfunctional tumour-infiltrating CD4^+^ T cells in the overall anti-tumour immune response [461].

Cytotoxic CD4^+^ T cells should be mentioned here. The principal difference between these cells and Th cells is their ability to kill infected and transformed cells i.e., their direct cytotoxic potential [386]. In human bladder cancer [462] has been demonstrated their capacity to kill autologous tumour cells in a direct way.

▪Pro-tumour immunity

CD4^+^ T cells are influenced by the TME, leading to adopt pro-tumour functions and secret factors that support the tumour survival. They may act in combination with other cell types, such as MDSCs, TAMs [431]. In addition to consumption of key nutrients (glucose, amino acid, arginine or tryptophan), accumulation of lactate and lactic acid can alter T cell expression of chemokine receptors. In addition, high levels of extracellular adenosine that can promote immune evasion [463].

Overexpression of IL-6 signalling pathway and the presence of TGF-β, correlating with immune evasion have a negative influence on Th1 cells. Although TGF-β has been associated with the inhibition of effector T cell by CD103^+^ upregulation, TGF-β-dependent CD103^+^ CD8^+^ subpopulation are long-lived tissue-resident memory cells with a potential cytotoxicity in epithelial tumours [386]. Additionally, due to its plasticity can be converted into Treg. Th1-Treg conversion is characteristic of both murine and human lung carcinoma [383] (Table 2), among others. Though, it was found that oxygen-sensing prolyl-hydroxylase (PHD) could block this conversion preventing melanoma metastasis to lung [464]. In addition, although Th1-derived IFN-γ is related to anti-tumour activity, IFN-γ can also have pro-tumour effects since cancer cells may express PD-L1 after exposure to IFN-γ [386].

Th2 cells have been mostly related to pro-tumour activities. They act by enhancing angiogenesis and by inhibiting cell-mediated immunity and subsequent killing of tumour cells. There are key factors that are involved in Th2-mediated immunity that can play a pro-tumoural role, such as eosinophils (IL-10 inhibits tumour cell lysis by CTLs); IL-13 (NKT-cell-derived IL-13 inhibits CTL activation); Type 2 CD8^+^ T cells (Tc2 cells by IL-10 secretion); B cells (inhibit CTL-mediated tumour clearance by IL-10 secretion and formation of immune complexes) [435]. Cancers such as human [465] and mice [213] breast cancer, human pancreatic [466] and human colon cancer [385] are evidence of Th2 cell pro-tumour activity (Table 2).

Th17 cells, have also a dual effect, since IL17A/F can promote angiogenesis in a direct (on tumour cells) or indirect (on stromal cells of the TME) manner [441]. Th17 has been related to bad prognosis in several cancers such as pancreatic [467], hepatocellular [468], colorectal [469] and hormone resistant prostate carcinoma patients [470]. It is able to induce angiogenesis in human cancers such as colorectal [389] (Table 2), gastric [471], hepatocellular [468], lung [472], pancreatic [467], and breast tumours [473]. In addition to that, this pleiotropic cytokine can promote tumour metastasis, inhibit tumour cell apoptosis and hinder proper anti-tumour responses [386]. Furthermore, in mouse models, Th17 cells and hence, IL-17 had the capacity to promote the development, the function and the recruitment within tumours of the MDSCs [474].

Another cell with paradoxical functions is Th9 cell. Given that IL-9 is a crucial cytokine involved in regulating the function of Treg cells and mast cells [452], in some contexts, when it is overexpressed, favours tumour growth [386], such as in Hodgkin lymphoma [475], anaplastic large-cell lymphoma [476], adult T-cell leukaemia [477], NKT cell lymphoma [478] and chronic lymphocytic leukaemia [479]. In addition, studies have also associated Th9 with bad prognosis in Cutaneous T cell Lymphoma patients [480], human hepatocellular carcinoma [481] and human lung cancer [387] (Table 2).

Tfh cells were negatively related to survival in a mouse model of hepatocellular carcinoma by Inflammation-induced IgA^+^ cells [373] (Table 2). Moreover, Tfh cells might have protective roles in non-lymphoid tumours; nevertheless, they have been related to bad prognosis when they are localised in neoplastic follicles in lymphoid tumours. Of note, in lymphoid tumours, non-neoplastic Tfh cell infiltrates have a better prognostic that neoplastic Th [457]. Hence, angioimmunoblastic T-cell lymphoma [482], CLL [483] and follicular lymphomas [484] showed cells that resemble the Tfr cells, which has been associated with bad prognosis.

##### Regulatory T Cells

Unlike conventional the conventional CD4^+^ effector lineage, regulatory T cells (Treg), another type of Th subset, they are characterised by immunosuppressive activity and tolerogenic functions in both, homoeostasis and inflammation. Being considered immunosuppressive in nature, they do not develop anti-tumour activity per se, as conventional T cells do. Several preclinical animal models have shown its critical role in tumour immunity, as only its depletion (by anti-CD25 depleting antibody or CD4 depletion in mice) or conversion into effector T cell can contribute to anti-tumour immunity, preventing tumour growth.

The interaction of chemokines with their receptors (CCR4 with CCL17 or CCL22; CCR10 with CCL28, and CXCR4 with CXCL1) are factors that influence their recruitment into TILs [485,486].

They produce inhibitory cytokines (IL-10 and TGF-β) and can be divided into two groups depending on where they are synthesised: induced Treg (iTreg) in periphery and thymic Treg (tTreg), in the thymus [485].

They were considered a support for the TME survival since they can suppress anti-tumour immune effector responses in the TME both by IL-10 and TGF-β production [486]. Hence, Treg immunosuppressive role has been described in giloblastoma patients (tumour recurrence and poor prognosis) [392] and in CRC patients [393], inhibiting effector T cells (Table 2). Furthermore, a mechanism that tumour cells can adopt to evade the immune system is CD4^+^ T cell conversion, i.e., effector CD4^+^ T cells and naïve CD4^+^ T cells can be modulated by the milieu of the TME and be converted into Tregs or iTregs, respectively [487]. For instance, a study published in 2011, demonstrated that gastric cancer cells were able to induce the conversion of CD4^+^ naïve T cells to Treg by TGF-β1 [488,489]. In addition, other mechanisms such as the differentiation resulting from the suppression of APCs by molecules in the TME or the expansion as a result of DC stimulation, could also lead to obtain Treg in TME [490]. For these reasons, Treg depletion has been related to good prognosis in cancer patients [491,492]. Notwithstanding, as mentioned above, they may indirectly contribute to anti-tumour immunity by inducing the reverse process, i.e., by converting Treg into anti-tumour effector CD4^+^ T cells, as recently suggested in patients with metastatic melanoma, gastrointestinal, and ovarian cancer [391] (Table 2).

Thus, despite its immunosuppressive function, some studies have shown that Treg fragility can be induced by the loss or mutation of Foxo1 from mature Tregs leading to Treg instability of intratumoural Treg within TME and consequently promote anti-tumour immunity [493,494].

##### CD8^+^ T Cells (Cytotoxic T Lymphocytes, or CTLs)

▪Anti-tumour immunity

CD8^+^ T cells are the major effectors in the anti-tumour immune response [495], being considered important in tumour surveillance. Upon specific recognition of peptide via MHC-I on APCs, naïve CD8^+^ T cells are activated and then differentiated into cytotoxic effector T cells. They execute effector function by a cytotoxic process (transient cell–cell interaction and directed release of cytotoxic proteins stored in lytic granules inducing DNA fragmentation) that lead to the death of the target cell. Similarly to NK cells, CD8^+^T cells can bind to and attack more than one target cell sequentially [496]. Hence, their ability to kill tumour cells make them plays a pivotal role in anti-tumour immunity and thus, to be highly relevant in the clinical control of cancer development and progression. Several studies have shown the need of a certain level of expression of MHC-I by the tumour to trigger a T-cell response through the TCR complex. Supporting this, it was observed that tumour-specific T cells in combination with IL-2 obtained complete neoplasm eradication [245].

Recently, a potent tumour suppression mechanism has been associated with CD8^+^ T cells recruitment by the transcription factor GATA Binding Protein 4 (GATA4) [497]. Although it was thought that CD8^+^ T cells were a homogenous group of cytotoxic cells that produce IFN-γ; nowadays, it is well-known that there are multiple subsets of CD8^+^ T cells (Tc1,Tc2,Tc9,Tc17,Tc22), not all of which (Tc9 and Tc17) have cytotoxic function [394].

Tumour-derived chemokines such as CXCL9, CXCL10, and CXCL11 are attracted by CXCR3 (expressed on activated CD8^+^ T cells), leading to their recruitment and development of anti-tumour immune response. The presence of cytokines IL-2, IL-7, IL-15 and IL-21 also promote CD8^+^ T cell activation [490].

Tc1 cells (characterised by IFN-γ, TNF-α production) are considered as the most frequent subset present in different cancers and correlating, in some cases, to favourable prognosis (melanoma, ovarian, breast, and lung cancer) [394] (Table 2). Their anti-tumour immunity is expressed via their exceptional cytotoxic potential in conjunction with their capacity to produce high IFN-γ levels [498], thus correlating with the upregulation of MHC-I on APCs [499]. IFN-γ is also capable of preventing the evasion of immunity by tumours, since it has the capacity to reprogram the suppressive cells in the TME, such as Treg cells [493].

Tc22 cells (characterised by IL-22, TNF-α, IL-2 production) were associated with a potential improvement with prognostic significance in human ovarian cancer through IL-6 induction [394] (Table 2). Tc17 cells (characterised by IL-17A, IL-17F, IL-21, IL-22 production and low levels of granzyme B) [394] have been associated with both anti-tumour (esophageal SCC patients) [396] and pro-tumour activity (HNSCC patients) [397] (Table 2).

Memory CD8^+^ T cells are the ultimate barrier of cancer immunotherapy due to its capacity to protect from recurrence and relapse [278]. It is well-known that CD103^+^ cDC1 cells are capable of leading to memory CD8^+^ T cell responses after CD8^+^ T cell priming, as reported in melanoma model [500]. In accordance with this, tumour infiltrated CD8^+^CD103^+^ T_RM_ cells (resident memory CD8^+^ T cells) has been associated with prolonged survival and better prognosis in ovarian [501], endometrial [502], breast [503] and lung cancer patients [504]. Nonetheless, memory CD8^+^ T cells often exhibit dysfunctional phenotypes and their dysfunction correlates with cancer progression [505].

Moreover, γδ T cells, a subgroup of T cells that belongs to the non-conventional or innate lymphocyte family, are also among the T cell sub-populations found in tumours. Activation of γδ T cells is mediated by a range of cellular and molecular determinants, including tumour-derived stress ligands and cytokine signals. They play a pivotal role in in the immunosurveillance of tumours and are used in cancer immunotherapies as well, owing to their ability to respond early (within seconds to minutes, rather than days) [506] to oncogenic transformation without needing clonal expansion. They can kill tumour cells, recruit other immune cells and, produce inflammatory mediators such as IFN-γ and TNF-α and bridge the innate and adaptive immune systems [506,507]. Their anti-tumour effect has been described in human acute myeloid leukemia and human breast cancer, among others [400] (Table 2).

It should be emphasised that, similarly to NK cells [508], as mentioned above, the functional activity TILs T cells depends also on stage tumour [509,510]. In accordance with all these reports provided, recently, the pivotal importance of the lymphocytes in the tumour was reviewed. In this review, the presence of several T cells-subtype (CD4^+^, CD8^+^, CD3^+^ majorly) was shown, both within the tumour as well as surrounding the tumour, by T-lymphocyte-based immunoscore (a novel prognostic tool focused on the presence of several T cells). In hepatocellular carcinoma, colon, melanoma, lung, pancreatic, breast cancer and even brain metastases, the presence of T cells has been associated with a good prognosis [511].

▪Pro-tumour immunity

The function of T cells is dependent on their interaction with tumour cells and other TME cells, thereby, the factors influencing its activation include (apart from metabolic factors) TGF-β, IL-10 and ROS (by TAMs); Arg1, iNOS and ROS (by MDSCs); PD-L1 or CD80 receptors expression (in tumour cells) [490].

Low Tc1/Tc2 ratios (along with imbalance of Th1/Th2) have been correlated with poor prognosis in patients with salivary gland tumours. Moreover, the increase in IL-22 by Tc22 has been positively associated with pro-tumour activity since it is capable of inducing tumour growth in transplant-associated SSC patients [399] (Table 2).

High Tc17s in the TILs has been correlated with a poor prognosis in human cancers, such as hepatocellular carcinoma [396], gastric [398], lung [512] and HNSCC cancer [397] (Table 2). Although the causes of these findings are not clear yet, it has been suggested that is likely, in part, due to IL-17 production by Tc17 cells, given that this cytokine has the ability to induce the production of inhibitory and angiogenic factors (such as VEGF) and the recruitment of cells with pro-tumourigenic capacity (such as neutrophils and MDSCs) [394].

In terms of survival and relapse [513], the γδ T cell infiltration level in breast cancer was crucial to indicate the state of the disease, being associated with immunosuppressive functions (pro-tumour role in various human malignancies), such as inhibition of naïve T cell proliferation and the impairment of DCs maturation and function [514]. γδ T cells subsets were also negatively involved in human cancers such as ovarian [515], PDA [516], rectal [517], and gallbladder cancer [401] (Table 2). Similarly to NK cells, γδ T cells function was also inhibited by tumour-derived PGE2 influencing their cytolytic functions and linked to high tumour grade and metastatic dissemination [355]. A study showed that under determined conditions (accumulation and activation of IL-23-producing inflammatory DCs caused by tumour dysplasia) γδ T cells were identified as the main cellular source of IL-17 in human CRC [518].

Another mechanism influenced and modulated by TME is the “exhaustion of CD8^+^ T cells”. This term is referring to special hypo-responsive state caused by persistent antigen stimulation in chronic virus infections or tumours that may be subjected T cells. This process gives rise to progressive loss of effector function by impaired self-tolerance mechanisms during thymic maturation leading to low affinity for antigen recognition by self/tumour specific T cells. The lack of innate stimulators causes APCs to be weakly activated giving rise to defectively activated tumour-specific T cells. This privation of effector function involves the loss of TNF-α, IFN-γ [519] and high levels of PD-1, as reported in several human cancers such as melanoma [395] (Table 2), NSCLC [520], HNSCC [521], gastric cancer [522] and ovarian cancer [523]. In addition to these features, the upregulation of immune checkpoints has been described as one of the hallmarks of T cell exhaustion [524]. Moreover, hyporesponsive state by T cells is not only maintained but also promoted by the TME. Thus, tumour-specific exhausted T cells are not able to control tumour progression in the late stage owing to immunotolerance and immunosuppression mechanisms present [519]. Despite all of these observations, it was demonstrated that tumour-specific T cell dysfunction is reversible with immune checkpoint blockade [525]. Previously, it was described that T cells with a hyporesponsive phenotype because of nutrient-poor conditions within their microenvironment cannot be reverted by subsequent stimulation [524]. More recently, it was observed that individually ineffective interactions by CD8^+^ T cells in non-hematologic solid tumours can improve at high CTL density, i.e., via “additive cytotoxicity” [496].

## 7. Concluding Remarks and Future Perspectives

Currently, it is well-known that the IS is able to kill both tumour and transformed cells through its anti-tumour immunity. However, through a process called “cancer immunoediting”, this ability can be compromised, as tumour cells can escape IS immunosurveilance, triggering tumour growth.

Previously it was believed that the tumour was harmful on its own; nevertheless, in order for the tumour to continue to grow and thus survive, it needs the “help” of different components, including immune cells, which play a key role; hence, they are of increasing interest.

Thus, TME, in an attempt to survive, educates immune cells with the aim that they cannot carry out their anti-tumour immunity. Primarily TME-derived factors, which in turn are influenced by factors such as tumour stage and type, are responsible for determining the function of immune cells.

Thereby, immune cells can be converted into immunosuppressive, innocent bystanders or even pro-tumour cells, thus showing the controversial effects of immune cells. In this context, TME and immune cells influence each other’s fates in a bidirectional manner. Accordingly, TME has become one of the most important research areas in cancer biology. A better understanding of the processes involved in immune evasion and the relationship between immune cells and TME may shed light on manipulating both immune cells and tumour fate.

## Figures and Tables

**Figure 1 cancers-14-01681-f001:**
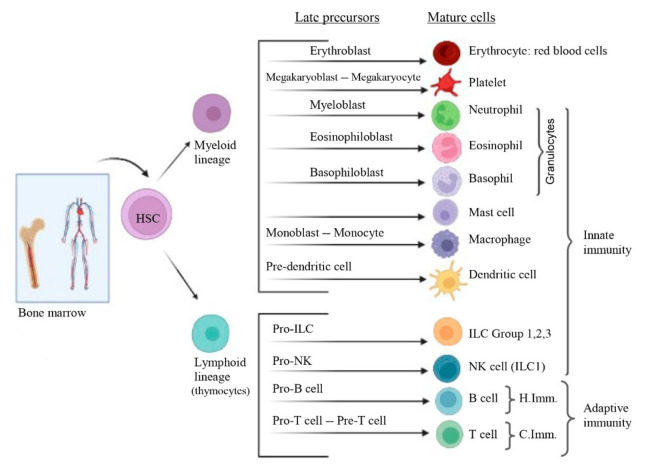
Hematopoiesis. The hematopoiesis process begins with the division of the hematopoietic stem cell (HSC), found in the bone marrow. This type of cell has the ability to be multipotent and self-renewing. HSCs, through the production of multipotent hematopoietic progenitor cells (HPCs) can give rise to two types of blood cell lines (myeloid and lymphoid lineage). Each of them, at the end of the process, will give rise to late precursors and mature cells forms through committed precursors. Mature cells are divided into three groups: red blood cells, platelets and leukocytes. Leukocytes form the innate and adaptive immunity, this latter subdivided in humoral (governed by B cells) and cellular immunity (by T cells). HSC: Hematopoietic stem cell; ILC: Innate Lymphoid Cell; NK cell: Natural Killer cell; H.Imm: Humoral immunity; C.Imm: Cellular immunity.

**Figure 2 cancers-14-01681-f002:**
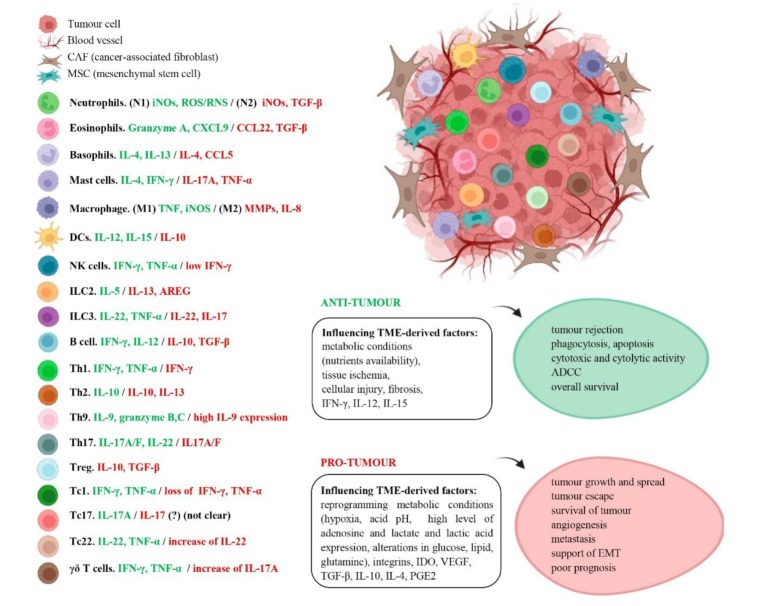
TME in anti- and pro-tumour conditions. The TME is a heterogeneous mass of tumour cells, stromal cells (such as MSC, CAF), blood vessels and immune cells. On the left, the cytokines secreted by the different types of immune cells are shown. Those released in anti-tumour activity are shown in green and those released in pro-tumour activity in red (in some cases the same cytokine has dual effect). On the other hand, on the right, TME-derived factors (which influence immune cell fate) in general, and their outcomes, accordingly, are shown, both in an anti-tumour and pro-tumour microenvironment. iNOS: inducible nitric oxide synthase; ROS/RNS: reactive oxygen species/reactive nitrogen species; TGF-β: transforming growth factor beta; CXCL9: C-X-C chemokine ligand 9; CCL22: C-C Motif Chemokine Ligand 22; IL-4: interleukin-4; IFN-γ: interferon gamma; TNF-α: tumour necrosis factor alfa; MMPs: matrix metalloproteinases; AREG: amphiregulin; TME: tumour microenvironment; ADCC: antibody-dependent cellular cytotoxicity; IDO: indoleamine 2,3-dioxygenase; VEGF: vascular endothelial growth factor F; PGE2: prostaglandin E2; EMT: epithelial-mesenchymal transition.

**Figure 3 cancers-14-01681-f003:**
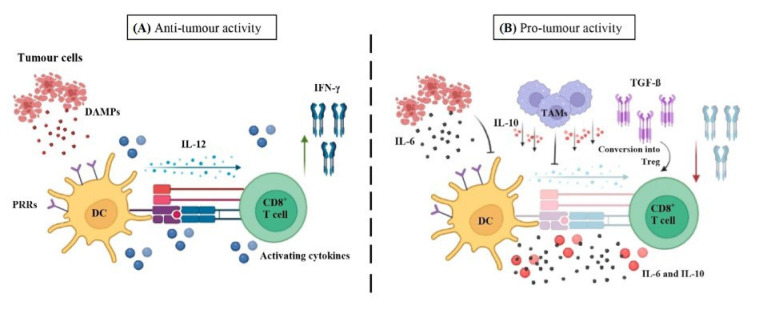
Anti- and pro-tumour microenvironment of DCs and CD8^+^ T cells. (**A**) Antitumour microenvironment: Antigen presentation by DCs via MHC-I to CD8^+^ T cells is depicted. For T cell activation to take place, apart from antigen presentation, co-stimulatory signals (DAMPs released by tumour cells, which are recognised by the PRRs of DCs and thus activated) and activating cytokines (represented by the blue round shapes) are necessary. Accordingly, activated DC can also promote CD8^+^ T cell by releasing IL-12 which induces IFN-γ production by CD8^+^ T cells. (**B**) Immunosuppressive microenvironment. However, factors such as tumour cells-derived IL-6 production or TAMs-derived IL-10 production (involved in inhibiting DC-derived IL-12) can negatively influence maturation, migration, differentiation and hence, function of DCs, downregulating them and give rise to a pro-tumour microenvironment. TGF-β, another factor present in TME, is able to convert effector T cells into Treg, creating an immunosuppressive environment. DAMPs: Damage-associated molecular patterns; PRRs: pattern recognition receptors; DC: dendritic cell; IL-12: Interleukin-12; IFN-γ (interferon gamma); TAMs: Tumour-associated macrophages.

**Table 1 cancers-14-01681-t001:** Innate immunity. Anti- and pro-tumor functions of the immune cells in different cancer types.

Immune Cell Type	Anti-Tumor Immunity	Pro-Tumor/Immunosuppressive Immunity	References
Granulocytes	Neutrophils (N1): promote T cell cytotoxic function in human lung cancer Eosinophils: strong cytotoxic activity in human colon carcinoma cells Basophils: involved in CD8^+^ T cell infiltration in murine colon cancer	Neutrophils (N2): promote M2 phenotype macrophages in mouse lung cancer Eosinophils: inhibition of T and NK cell function in human NSCLC Basophils: reduction of CD8^+^ T cell infiltration both human and murine colon cancer	[153]/[97] [154,155]/[156] [157]/[158]
Mast cells	Cytolytic activity in breast cancer patients	Immunosuppression of T cells in human gastric cancer	[159]/[160]
Macrophages	(M1): Apoptosis in xenograft mice pancreatic cancer	(M2): Th1 responses suppression in mice breast tumor cells	[161]/[162]
Dendritic cells	CD103^+^cDC1s: cross-presentation, trafficking tumor antigen to lymph nodes and enhancement of effector CD8^+^ T cells in hepatocellular carcinoma patients moDCs: capacity to scavenge tumor antigen in lung and CRC patients Langerhans cells: APC function in primary lung adenocarcinoma patients	CD103^+^cDC1s: impairment of anti-tumor CD8^+^ T-cell responses in human CRC moDCs: impairment of T cell activation in human SCC Langerhans cells: dysregulation of DOK via IL-1β in human OSCC cells	[163]/[164] [165]/[166] [98]/[167]
ILC Group 1	NK cells: elimination of tumor cells in leukaemia mouse model ILC1s cells: cytotoxicity through ILC1s-derived IFN-γ in early-stage of human multiple myeloma	NK cells: defect of cDC1 recruitment into the TME by NK cells in human gastric cancer cells ILC1s cells: inhibitors receptors expression at a late stage in CRC patients	[168]/[169] [170]/[171]
ILC Group 2	ILC2s: indirect support by enhancement of DCs and effector T cells in primary and metastatic murine lung	ILC2s: involved in MDSCs immunosuppressive function in human bladder cancer	[172]/[173]
ILC Group 3	ILC3s: contribute to tumor suppression by TLS, cytokines secretion and tumor cells recognition in human NSCLC	ILC3s: contribute to metastasis by secreting IL-17, IL-23 and IL-22 in human CRC	[174]/[174]

Abbreviations: N1,N2: N1/2-polarized neutrophils; M1,M2: M1/2-polarized macrophages; NK cells: natural killer cells; NSCLC: non-small-cell lung cancer; DC: dendritic cells; cDC1s: conventional dendritic cells type 1; CRC: colorectal cancer; moDCs: monocyte-derived dendritic cells; SSC: squamous cell carcinoma; APC: antigen-presenting cells; DOK: dysplastic oral keratinocytes; OSCC: oral squamous cell carcinoma; NK cells: natural killer cells; ILC1/2/3s: innate lymphoid cells 1/2/3; IFN-γ: interferon gamma; MDSCs: myeloid-derived suppressor cells; TLS: tertiary lymphoid structures; IL: interleukin.

**Table 2 cancers-14-01681-t002:** Adaptive immunity. Anti-and pro-tumor functions of the immune cells in different cancer types.

Immune Cell Type	Anti-Tumor Immunity	Pro-Tumor/Immunosuppressive Immunity	References
B cells	B cells: antibodies production in human lung cancer (anti-MUC1; anti-p53).	B cells: immune complexes in human OSCC Breg cells: disruption of Th1/Th2 balance in human gastric cancer	[371,376]/[370] [370]
CD4^+^ T cells	Th1: IFN-γ production in melanoma patients Th2: IL-10 production in primary human prostate cancer Th9: Granzyme production in human lung cancer Th17: induce apoptosis in human CRC Tfh: organisation of TLS in breast cancer patients Treg cells: only when subjected to conversion (to effector CD4^+^ T cells) as suggested in metastatic melanoma, gastrointestinal, and ovarian cancer patients	Th1: defective function in both murine and human lung carcinoma Th2: IL-4 production induces EMT in human colon cancer. Th9: tumor metastasis in human lung cancer Th17: angiogenesis in human CRC Tfh: IgA^+^ cells in mice hepatocellular carcinoma. Treg cells: inhibition of effector T cells in glioblastoma and CRC patients	[382]/[383] [384]/[385] [386]/[387] [388]/[389] [390]/[373] [391]/[392,393]
CD8^+^ T cells	Tc1: cytotoxic activity and high IFN-γ production in melanoma patients Tc17:IL-17A production in oesophageal SCC patients Tc22: through by IL-6 induction in human ovarian cancer γδ T cells: NKp30 cytotoxicity in human acute myeloid leukemia	Tc1: low of IFN-γ, TNF-α and high levels of PD-1, incapacity to control tumor progression in melanoma patients Tc17: angiogenic and immunosuppressive function in human cancers such as HNSCC and gastric cancer Tc22: increase of IL-22 related to tumor growth in transplant-associated SSC patients γδ T cells: angiogenesis by IL-17 production in human gallbladder cancer	[394]/[395] [396]/[397,398] [394]/[399] [400]/[401]

Abbreviations: MUC1: mucin 1; OSCC: oral squamous cell carcinoma; Breg cells: regulatory B cells; Th: T helper; Tfh: T follicular herlper; IFN-γ: interferon gamma; IL: interleukin; EMT: epithelial-mesenchymal transition; CRC: colorectal cancer; TLS: tertiary lymphoid structures; Treg cells: regulatory T cells; PD-1: programmed cell death protein 1; SCC: squamous cell carcinoma; HNSCC: head and neck squamous cell carcinoma; NKp30: natural killer cell protein 30 (number refer to its molecular weight).

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
