# Peer review of "Dual Effect of Immune Cells within Tumour Microenvironment: Pro- and Anti-Tumour Effects and Their Triggers"

_cancers, 2022, doi:10.3390/cancers14071681_

Round 1

Reviewer 1 Report

The review on Dual effect of immune cells within tumor-microenvironment is very comprehensive with an excellent background and recent advancements in this field.  Overall the manuscript is well written and organized; there are minor changes and suggestion which need to be done before this paper is ready for the publication

  1. This line needs to be rephrased and corrected: However, in several cases, tumour cells are able to evade the immune system. As well as immune cells, the extracellular matrix, blood vessels, fat cells and various molecules and cells support tumour growth and development.
  2. Line 18: grow needs to be replaced by growth
  3. This line needs to be rephrased and corrected: Edward Jenner, known as "the father of immunology" was the pioneer of this event and who got the World Health Organization (WHO) to announce the eradication of smallpox in 1979 [1].
  4. Figure 1: Peripherally blood needs to be replaced by peripheral blood
  5. Figure 2: Green oval text needs to be checked, I think tumor growth and spread should be moved to red oval shape
  6. Check font size line 505 to 511
  7. Tables 1 and 2, Immune cell heading can be replaced by Immune cell type

Reviewer 2 Report

In this review, the author introduced the basic knowledge of the immune system and the dual role of the immune system in cancer development. My overall expression of this review is that it includes too much information in the text, however, each point that the author summarized was too basic and simple.

Some major concerns:

  1. Neutrophil and Macrophage biology is my expertise. I went through the text in terms of the role of neutrophils and macrophages in cancer immunology. Some important findings are missing, for example, Neutrophils escort circulating tumor cells to regulate breast cancer;  DNA of neutrophil extracellular traps promotes cancer metastasis via CCDC25; FATP2 regulates the uptake of AA in neutrophils to regulate tumor growth; PI3Kr regulates macrophage polarization to control tumor growth........
  2. Although people define the role of macrophages and neutrophils in tumors by classifying macrophages and neutrophils into M1 M2 macrophages or N1 N2 neutrophils, this is a kind of artificial classification that is simply based on the cytokine profile or signaling pathway activation in different polarized cells. In the real tumor microenvironments, TAM phenotypes are much more complex and cannot be categorized into binary states. So the author needs to include this point in the text.

Minor point

  1. In Fig.1, the author stated: "The hematopoiesis process begins with the division of the hematopoietic stem cell, found in the bone marrow and the blood peripherally." In healthy humans or mice, HSC is rarely found in peripheral blood, the bone marrow is the major site for hematopoiesis.  Mobilization of HSC to peripheral blood or peripheral organ occurs when bone marrow fails to function normally, which leads to extramedullary hematopoiesis. In Fig.1, I think the author wants to describe normal hematopoiesis, so the peripheral blood needs to be removed.

Suggestions:

        1. My suggestion is that it is better to pick one or two cell types as the target to review. So the author can review more deeply than the current version.

Round 2

Reviewer 2 Report

The authors well addressed the questions and added some important findings to the manuscript. I am still concerned about the depth of each topic the author reviewed but since the author intends to give a broad picture of how immune cells are involved in cancer biology, the current version is good for publication.